# Behavioral Pedestrian Tracking Using a Camera and LiDAR Sensors on a Moving Vehicle

**DOI:** 10.3390/s19020391

**Published:** 2019-01-18

**Authors:** Martin Dimitrievski, Peter Veelaert, Wilfried Philips

**Affiliations:** TELIN-IPI, Ghent University - imec, St-Pietersnieuwstraat 41, B-9000 Gent, Belgium; martin.dimitrievski@ugent.be

**Keywords:** pedestrian tracking, multi-object tracking, particle filter, behavioral, sensor fusion, LiDAR, autonomous vehicle, driverless car

## Abstract

In this paper, we present a novel 2D–3D pedestrian tracker designed for applications in autonomous vehicles. The system operates on a tracking by detection principle and can track multiple pedestrians in complex urban traffic situations. By using a behavioral motion model and a non-parametric distribution as state model, we are able to accurately track unpredictable pedestrian motion in the presence of heavy occlusion. Tracking is performed independently, on the image and ground plane, in global, motion compensated coordinates. We employ Camera and LiDAR data fusion to solve the association problem where the optimal solution is found by matching 2D and 3D detections to tracks using a joint log-likelihood observation model. Each 2D–3D particle filter then updates their state from associated observations and a behavioral motion model. Each particle moves independently following the pedestrian motion parameters which we learned offline from an annotated training dataset. Temporal stability of the state variables is achieved by modeling each track as a Markov Decision Process with probabilistic state transition properties. A novel track management system then handles high level actions such as track creation, deletion and interaction. Using a probabilistic track score the track manager can cull false and ambiguous detections while updating tracks with detections from actual pedestrians. Our system is implemented on a GPU and exploits the massively parallelizable nature of particle filters. Due to the Markovian nature of our track representation, the system achieves real-time performance operating with a minimal memory footprint. Exhaustive and independent evaluation of our tracker was performed by the KITTI benchmark server, where it was tested against a wide variety of unknown pedestrian tracking situations. On this realistic benchmark, we outperform all published pedestrian trackers in a multitude of tracking metrics.

## 1. Introduction

As our society steadily embraces automation in our transportation systems, the demands from computer vision systems are increasing with an accelerated pace. This is especially true of driverless car prototypes which, seemingly, have difficulties navigating the complex traffic landscape. It is obvious that contemporary sensors cannot match the perception capabilities of a highly skilled and never tired human driver. However, there is also a fundamental problem in our understanding of how human drivers react in unpredictable situations. Traffic attitude, among many other human behavioral phenomena, can not easily be modeled with standard statistical tools. Currently, level 4 and 5 autonomy on public roads requires safety levels that are beyond the capability of any available system on the market. Transitioning to these levels of autonomy requires overlapping and redundant systems where the driving without human intervention must continue even in the event of a sensor failure. By utilizing multi-sensor fusion, algorithms in the literature such as object detection and tracking have made strides in terms of precision, robustness, and redundancy. However, the translation of such systems in the real world has been so far incomplete.

An additional hurdle is imposed by the ability for a perception system to equally perform across the spectrum traffic scenarios. These include the difficult types of environments and scenarios where one or several of the sensors may be overloaded, interfered or cluttered. In a nutshell, when road safety is concerned, we humans, as well as the autonomous driving systems, must rely on a combination of 2D and 3D sensing, well-defined traffic rules as well as non-written contextual cues to safely reach our destination. The latter is especially challenging to teach to a computer since traffic user behavior can be affected by a multitude of seemingly abstract stimuli. One can think of errors in positioning, detection, and missing sensor data as low-level noise, and unpredictable, and sometimes chaotic, behavior of road users as highly abstract noise patterns. In the global ecosystem of autonomous vehicles, there exist many other challenges such as vehicle control, path planning, communication and cooperation, ethical reasoning, etc., all of which are outside of the scope of this paper. Thus, we hereby discuss the environmental perception sub-system where the core problem consists of accurate tracking of objects that interact with our ego vehicle.

In the specific context of this paper, we define environmental perception as the process of extracting knowledge about obstacle positions, velocities and intentions from sequential noisy observations. We follow the paradigm of online tracking-by-detection, where a Convolutional Neural Network (CNN) detects candidate observations which are then assigned to pedestrian models using the Bayesian formulation and Sequential Monte Carlo simulations. Image detections are fused by additional range information from a 3D LiDAR sensor in a joint log-likelihood observation model. Thus, our tracker makes the best use of the available imaging and ranging data to solve for ambiguous situations, e.g., occlusion and bad lighting. The motion of tracked pedestrians is modeled by behavioral motion models that closely match short-term human behavior in general traffic situations. These behavioral models drive the initialization and update steps in the Monte Carlo simulations using statistics from offline training data. We represent each pedestrian position and velocity by sample-based non-parametric probability distributions that allow for great flexibility in modeling uncertainties. High-level track actions are handled by a track management system which decides to create, delete, update or merge tracks based on a probabilistic track score. Last but not least, we estimate vehicle odometry from the range data itself which is then used to perform tracking in a global coordinate system. Thus, our tracker can track multiple pedestrians from a moving vehicle in various traffic situations which we confirm by independent evaluation. We report the best performance among all currently published methods in most tracking metrics on one of the most widely used 2D–3D tracking benchmark for autonomous vehicles, [1].

In the following section, we present a wide overview of the relevant tracking literature. In Section 3, we formulate the problem and we summarize the novelties of our work. Then, in Section 4, we give a detailed explanation of the employed particle filter tracker, followed by our motion and observation likelihood model definitions in Section 5. In Section 6, we present our data association and track management framework, and, finally, in Section 7 we give the experimental methodology with our quantitative and qualitative findings. We conclude the paper with closing remarks on some of the fail cases and directions for future improvement.

## 2. Related Work

Multi-Object Tracking (MOT) estimates the states of objects in a camera image by means of modeling the spatiotemporal interaction between the object and the background. Based on their problem definition, data input and timing requirements, tracking methods can be split into several categories. Methods based on foreground/background modeling abound in situations where the environment and the camera are generally static. In dynamically changing environments where the camera is in motion, MOT methods usually rely on sophisticated detectors to initialize and update the tracks. Temporally, methods can be split into ones that operate in the present, i.e., online, and ones that process historic data offline. Online methods are suitable for time-critical applications, while offline methods generally offer increased tracking accuracy. Offline methods take into account all past and future observations to estimate the state of each object at each time instant. There are of course a myriad of methods that lie somewhere in between, with approaches using cameras that rotate/zoom but do not change position, or trackers that are initialized with objects only in the first frame. There also exist so-called “near online” methods that introduce small time lag by using temporal windowing. In recent years, as discussed in the introduction, a trend in the published literature is to exploit additional modalities such as multi-spectral, range and/or positional data. Generally speaking, multi-modal object detection becomes simpler, when, for example, cues from range sensors can help to discriminate occlusions, or thermal signatures can lead to improved traceability in difficult lighting conditions. Last but not least, tracking of objects from a moving camera must be performed in a coordinate system that is detached from the moving camera. Positional information such as GPS, accelerometers or visual odometry provide the necessary ego vehicle data that can further be shared between other road users that are nearby.

Tracking research tries to increase tracking performance by innovations in several key areas: tuning object detection for tracking, assignment problem optimization, motion modeling, better state representation features, and, lately, the design of end-to-end CNNs. Finding globally optimal, solutions to the data association problem [2] has mainly been done using graphical models for connecting individual object detections into a consistent set of trajectories, with approaches offering a novel solution using different techniques such as k-shortest paths in DP NMS [3] or a Conditional Random Field as in DCO X [4] or a variational Bayesian model in OVBT [5]. Modeling the motion of targets within the image was also given a lot of attention with some of the successful approaches: SMOT [6] and CEM [7] and MotiCon [8]. Authors in these papers based the matching costs for comparing pairs of detections on simple distances and weak appearance models. These methods currently score around 10% worse than the state of the art. Very recently, there was a shift towards designing a strong appearance based similarity metrics for the pairwise matching. Some of the recent best performing approaches are based on sparse appearance models such as LINF1 [9] or online appearance updates in MHT DAM [10] and channel feature appearance models, oICF [11] and aggregated local flow of long-term interest point trajectories in NOMT [12]. In addition to the image based analysis, authors have proposed to exploit depth information in order to improve tracking performance with one of the most recent notable advances using a combined 2D–3D Kalman filter by [13].

Another notable trend is the proliferation of deep learning into the tracking community with sparse but notable examples such as MDPNN16 [14], which uses Recurrent Neural Networks in order to encode appearance, motion, and interactions. Another example is JMC [15] which uses deep matching to improve the affinity measure. There usually is a correlation between strong affinity models and tracking performance, which, together with machine learning approaches, is believed to be one of the key aspects to be addressed to further improve performance [2].

### 2.1. Zero Velocity 2D–3D Trackers

Zero velocity methods which use a rather simplistic model of the object motion. In some realistic scenarios, this motion model doesn’t correspond well to a person motion where trajectory jumping between persons or assignment of a wrong detection can create a typical fail situation. This is so because the movement and the direction are not captured by the trajectory representation. Additionally, these approaches operate as a first-order Markov Process in a sense that both their motion and appearance models are made from the observations in the previous frame only. A typical failure can happen when a person is lost for one instance and the tracking is automatically discontinued. Thus, a fresh trajectory must be created as soon the person re-appears in the sensor data regardless of how long he/she was missing.

A stereo-based tracking approach is presented in [16] based on the tracklet’s position and size constancy from frame to frame. Candidate objects are detected by a segmentation method that identifies connected components in the disparity images. These segments match the regions in the 3D space with a typical volume occupied by a person facing the camera. Faces and bodies of people are tracked over several temporal scales: short-term (user stays within the field of view), medium-term (user exits/reenters within minutes), and long-term (user returns after hours or days). Short-term tracking is performed using simple region position and size correspondences, while medium- and long-term tracking are based on statistics of user appearance. These features are used to solve occlusions and target re-identification in case of targets re-entering in the scene.

Authors in [17] use foreground Regions of Interest (ROI) to detect and track objects. Using the confidence score from a Histogram of Oriented Gradients (HOG) based detector, they select trajectories that, at some point of time, have high values. The matching stage is performed by dynamically building a directional graph of all detections. Trajectories representing strongly connected paths in the graph are extracted based on optimal cost and are treated as tracks and a greedy algorithm is used for extracting individual paths. The edge cost used for matching is estimated from the similarity of the color signatures measured using the Earth mover’s distance. Their trajectory representation consists of a zero velocity motion model and color signature appearance model.

Following an ROI selection stage, authors in [18] detect pedestrians using a combination of depth cues and a HOG detector. Detections are matched with trajectories from the previous frame by image patch-correlation. This is performed at the positions in the image that correspond to the previous observation of the person after correction for camera motion estimated by visual odometry. Each target trajectory representation is modeled by a zero-velocity motion model in the 2D image coordinates and an appearance model that consists of an image patch around the detection in the previous frame.

Dan et al. [19] use both top-view vision and depth images which are captured by a video and depth camera mounted on the ceiling. Objects are detected using a human model from the preprocessed depth image and every detection is then matched independently to detections in the previous frame. The matching maximizes a score that leverages appearance similarity and closeness in 3D space. The trajectory representation used for matching consists of RGB-D based dynamic appearance model with a sliding window of one frame and a zero-velocity motion model. A bidirectional matching strategy is used, where all detections in frame *t* are matched to those in frame t−1 and vice versa. This allows handling trajectory splits and merges that may arise from the failure of detection in one direction.

### 2.2. Constant Velocity 2D–3D Trackers

The following 2D–3D tracking approaches propose motion models using first order linear kinematics. Most of them also build the appearance model incrementally over time. Doing so allows for maintaining consistent trajectory representations, and helps to prevent the model from changing dramatically in cases of temporary detection failure over multiple frames. The problem of such approaches lies in the difficult cases of pedestrian motion, i.e., a person can move in unpredictable directions with changing velocity. Constant velocity models don’t have the capacity to explain such scenarios and often fail if the prediction window is long and unpredictable motion becomes apparent.

In the work of Han et al. [20], the tracklet’s motion is modeled by the mean and variance of its depth variations in the past ten frames. The appearance model consists of color and texture histograms for the torso and legs generated at the first instance of a new person. Trajectory representation is kept after the person leaves the scene in order to allow re-identification in case of re-entry. Trajectory matching is performed by selection from a linear combination of the appearance similarity and the continuity of the depth variation. Appearance matching is performed using with the Bhattacharyya distance measure and the depth variation is expected to follow a Gaussian distribution with a mean and variance provided by the motion model under the assumption of a constant speed.

The approach in [21] assumes a target velocity of 2 ms${}^{-1}$ in any one direction, which means that the motion model does not directly depend on the data. The trajectory appearance model consists of the color histogram of the last observation for the track and matching is performed by comparing candidate detections from depth information in ROI to trajectories based on the color histograms of the candidate and of the appearance model of the track. Only trajectories that are predicted to be located close to the candidates are considered as valid.

### 2.3. Kalman Filter Based 2D–3D Trackers

These approaches use the Bayesian theorem to model pedestrian states from current observations and the state variables in the previous frame. The state representation follows a Gaussian distribution, while motion is usually linear. The position and the velocity of the next observation are predicted from the model and then compared with the positions of new detections during the matching stage. Kalman trackers have a well-founded solution and are especially effective in tracking objects that do not change their behavior of motion.

Authors in [22] detect people in ROI using a cascade of RGB and depth-based detectors. Detected candidates from depth cues are verified by the HOG detector and by a poselet-based human detector that detects body parts. The matching stage locates the best matching track or static background object for each new detected candidate using a Directed Acyclic Graph (DAG) to handle the decision process. The DAG performs coarse matching by position similarity first and then finer matching to account for appearance similarity. A dynamic model is used to represent the appearance which is updated online by an AdaBoost algorithm.

Authors in [23,24] introduce the notion of Point Ensemble Image, which fully encodes both RGB and depth information from a virtual plan-view perspective. They reveal that human detection and tracking in 3D space can be performed very effectively based on their proposed representation. Next, they detect all candidate people in ROI of a new frame from RGB-D data and then, for each track, select the best detected candidate. The appearance model of the trajectory representation is a joint color and height histogram.

Authors in [25] propose a person identification method for a mobile robot which performs a specific person following under dynamic complicated environments. They detect people by applying a classifier cascade to the RGB-D data. An extended Kalman filter is used to track the target in the 3D space. The appearance model consists of SIFT features which are periodically collected from the target. The association between tracked targets and current frame detections is performed by thresholding on the number of matching SIFT features.

Another system, able to visually detect and track multiple persons using a stereo camera placed at an under-head position, is presented in [26]. This method detects people from a face detector applied in ROI selected from depth information. The matching stage finds the globally optimal associations of detected candidates to existing tracks using the Munkres (Hungarian) method. Matching likelihoods are computed from the distance to the predicted position and the similarity to the color histogram appearance model estimated with the Bhattacharyya measure. The appearance model is updated by linear combination of its current values and the new observed color data.

Authors in [27] exploit the Hungarian algorithm for matching detected and tracked objects, where they identify background areas with a depth-based occupancy grid system. Candidate targets are searched from foreground areas which is analyzed with a cascade of classifiers, comprising face and skin detectors and a full body HOG-based human detector. Detected objects are tracked simultaneously with a compressive tracker and a Kalman filter.

Harville [28] detects moving candidates by applying the background subtraction algorithm to RGB-D data. The detected foreground objects are projected to a 2D reference plane where occupancy and height maps are computed. Their Kalman filter tracking state includes position in the reference plane and the height and occupancy maps data. These features are linearly combined to calculate a score which is used in the measurements and updates of a Kalman filter.

Kalman filters have been studied in great detail and can produce satisfactory tracking results in many object tracking situations. However, we find this model to be less than ideal for tracking objects with uncertain behavior, especially in low frame rate sequential systems.

### 2.4. Particle Filter Based 2D–3D Trackers

A branch of MOT in the literature employs Particle Filters or Sequential Monte Carlo algorithms to do the state estimation of pedestrians. Particle filters represent the posterior distribution of a state variable using a set of samples or particles. A sequential re-sampling and mutation of particles is applied during the update–prediction process which can be nonlinear. A potential drawback of these approaches is that, in order to make a feasible approximation, particle filters can become computationally expensive. Authors usually apply various simplifications and optimizations to achieve practical running times.

Authors in [29,30,31] use a single particle filter per track. They use a constant speed model to predict the next location of the target and new target observations are located by maximizing a detection probability. Specifically, in [29,30], candidate objects are identified from ROI based on depth information and the probability of the presence of a person is computed based on the number of points in a cluster and its maximal height. To compute the probability of detecting the tracked person, this human presence probability is combined with an interaction factor that allows handling trajectory crossings by imposing a minimal separation between the positions of different people. In [29], the detection probability also includes the Bhattacharyya appearance similarity measure, while, in [30], it uses a measure of confidence on depth. Hence, the trajectory representation in [30] does not include any appearance model, and, in [29], it models appearance by the color histogram of the cluster. This model is updated with new observations that have high detection and matching confidence by the linear combination of the previous model and of the new histogram. In [31], the detection probability is made up of three terms. It includes the probability of being a frontal-facing human, firstly by verifying that the cluster may be approximated by a vertical plane at the expected distance from the camera, and secondly, by evaluating the fitting of an ellipse on the RGB image in order to validate the presence of the elliptical shape of a head at this position. It also uses the Bhattacharyya appearance similarity measure to compare to the trajectory representation’s appearance model, made up of two color histograms inside two ellipses of predefined sizes and respective positions that represent the head and torso, respectively. This appearance model is updated dynamically as in [29]. In all three methods, new tracks are initialized when unknown targets are detected based on the use of generic person descriptions. Tracks are kept for a number of frames after occlusion or departure.

Choi et al. [32,33] use particle filtering with Reversible Jump Markov Chain Monte Carlo (RJ-MCMC) sampling to track multiple people simultaneously, as well as static non-human objects (obstacles). Given the positions and velocities of all tracked targets and the results from generic person detectors applied to ROI, at each iteration, a move is attempted to initialize, delete or update a trajectory. Each move is sampled from a space of possible moves and a likelihood for the new solution is estimated. Moves are accepted or rejected similar to MCMC sampling until the chain converges. The moves are guided by the probability of continuous tracking, based on a smooth target’s motion constraint, which may also account for people interactions and the probability of being a human, while, in [33], authors account for the person’s appearance in the likelihood by computing the distance from a target-specific appearance-based mean-shift tracker, in [32], they do not use any appearance model and in [12] they define a novel Aggregated Local Flow Descriptor (ALFD) that encodes the relative motion pattern between a pair of temporally distant detections using long-term interest point trajectories (IPTs). Another contribution in [12] is a near online tracking approach using data-association between targets and detections in a temporal window, which is performed repeatedly at every frame.

In summary, particle filters exploit the paradigm of genetic algorithms in order to re-sample particles according to a fitness function. Therefore, the number of samples and how (and when) these samples are re-sampled can have a huge influence in the tracking performance. In the context of pedestrian tracking, to the best of our knowledge, particle filters pose the largest flexibility while having the least amount of downsides. Thus, they are the optimal candidates for building a pedestrian tracking system.

## 3. System Overview

### 3.1. General Layout and Contributions

A weakly calibrated camera-LiDAR pair, located on top of a moving vehicle, provides the system with observations. Prior to tracking, vehicle orientation to the road is estimated from the LiDAR data using our algorithm in [34] and image optical flow fields are computed using [35]. All tracking steps are performed in absolute coordinates by first registering current measurements. Object detection is done on a frame by frame basis using only the camera frames. A state-of-the-art CNN runs over each new camera frame and point cloud segmentation extracts 3D non-ground plane regions in the current LiDAR data. Image bounding boxes are then matched to segments in the LiDAR point cloud. We employ a novel fusion scheme where data is first processed at the sensor level separately, Figure 1 left, and matched 2D–3D observation features are compared to fused 2D–3D track estimates, assignment in Figure 1. Tracking of pedestrians is performed by dual 2D–3D particle filters operating on the prediction and update principle. We avoid completely fusing the 2D and 3D data into a single tracking state due to the less than ideal camera-LiDAR calibration. We find that current sensor technology is not mature enough to provide pixel accurate depth correspondences between image bounding boxes and LiDAR objects. Therefore, we specifically designed our tracker logic to cope with mis-calibration errors by allowing the 2D and 3D state estimates to move independently of each other. In this paper, we claim the following contributions to 2D–3D pedestrian tracking:

**Contribution I:** we propose a novel pedestrian motion model which, employed in a particle filter, has the capacity to track unpredictable behavior. On the ground plane, we use probabilistic longitudinal and lateral acceleration model from which we sample new velocities for each particle, illustrated in Figure 2. On the image plane, we propose a motion model with adaptive longitudinal and lateral acceleration conditioned on the EGO vehicle speed. Image particles change their velocities in proportion to how far they are from the camera and how fast the vehicle is moving. Details of this work are presented in Section 5.1 and Section 5.2.

**Contribution II:** a novel approach to separating data association from the state estimation where we perform tracking in separate 2D and 3D spaces, while data association is done in fused 2D–3D space. Specifically, our tracking model consists of independent image and ground plane state variables, illustrated in Figure 3, while the observation likelihood model is defined in joint 2D–3D feature space. The detailed formulation of our particle filter tracker is given in Section 4 and the observation likelihood model in Section 5.3.

**Contribution III:** a novel track management system for handling track existence and interactions based on the track confidence score. In this paper, we introduce bounds on the probabilistic track score function from [10,36] in order to increase the responsiveness to new evidence. Our bounded function produces higher tracking accuracy in a large set of traffic situations. We refer the reader to Section 6.3 for more details.

We are interested in knowing the pedestrian’s identity, position, and velocity from a set of noisy observations over time. Since our system operates on the tracking by detection principle, we will hereby define the observation and tracking state representations.

**Observation vector:** at time instance *t*, the sensors capture a pair of RGB camera image $I_{t}$ and a LiDAR point cloud $PC_{t}$ which are processed with an object detector and point cloud segmentation pipeline, Figure 1. Throughout this paper, we will refer to these detected objects as observation data. Each detection consists of 2D image plane position which is matched to an object in the PC. Formally, the CNN object detector produces a set of *n* detections in the image plane: $d_{i}^{image}\in \left\{d_{1},d_{2},\ldots ,d_{n}\right\}\left|n\geq 0\right.$ where each detection $d_{i}^{image}$ consists of the center, bounding box size and the detection score: $d_{i}^{image}:\left(r_{i},c_{i},w_{i},h_{i},s_{i}\right)$. We augment this image information by matching each detection $d_{i}^{image}$ to its ground plane location using the segmented LiDAR data. We additionally compute the image plane motion vector and an appearance feature for the top and bottom half of the bounding box. Thus, the complete observation vector $d_{i}$ consists of several 2D and 3D features:(1)$$\begin{array}{c} d_{i}:\left\{d_{i}^{image},d_{i}^{ground}\right\},\\ d_{i}:\left\{r_{i},c_{i},w_{i},h_{i},▵r_{j},▵c_{j},app_{i},x_{i},y_{i},s_{i}\right\}. \end{array}$$

### 3.2. Problem Definition

**State vector:** at time *t*, the tracker keeps a set of *m* targets $k_{j}\left|{}_{j=0..m-1}\right.$ which represent models of the true pedestrians. Each model forms a probability density function about the state of the person for which position, appearance and behavior data has been accumulated from past observations. Each target $k_{j}\in \left\{k_{1},k_{2},\ldots ,k_{m}\right\}\left|m\geq 0\right.$ consists of the following 2D and 3D state variables: in the image plane, we use the bounding box (BB) position estimate$\left(r_{j},c_{j},w_{j},h_{j}\right)$ and BB velocity estimate$\left(▵r_{j},▵c_{j}\right)$. From these state variables we can compute an appearance model $app_{j}^{top},app_{j}^{bottom}$ for values $r^{top}\in \left[r_{j}-\frac{h_{j}}{2},r_{j}\right)$ and $r^{bottom}\in \left[r_{j},r_{j}+\frac{h_{j}}{2}\right)$. The proposed appearance model $app_{j}\left(I,r_{j},c_{j},w_{j},h_{j}\right)$ loosely represents the visual appearance of the person (clothes, accessories, skin color, hair, etc.) in the image as defined by the tracklet bounding box, i.e., the color histograms. In 3D, the tracker state variables consists of the ground plane position and walking velocity $\left(x_{j},y_{j},▵x_{j},▵y_{j}\right)$. From these tracking states and observational evidence we can compute and update a track confidence score $\chi_{j}\approx p\left(d_{j_{1:t}}\left|j_{1:t}\subseteq k_{i}\right.\right)$, which gives the probability that the associated observations belong to a true pedestrian versus some background object. In other words, the track score models the overall spatio-temporal quality of how this pedestrian was tracked over time. Thus, for each pedestrian, the state variable vector $k_{j}$ is defined as:(2)$$\begin{array}{c} k_{j}:\left\{k_{j}^{image},k_{j}^{ground}\right\},\\ k_{j}:\left\{r_{j},c_{j},▵r_{j},▵c_{j},w_{j},h_{j},app_{j},x_{j},y_{j},▵x_{j},▵y_{j}\right\}. \end{array}$$

Internally, two sets of $N_{p}$ 8-dimensional particles are stored for each person which are used to model the non-parametric distributions of the image and ground plane positions and velocities while bounding box size and appearance are modeled with a simpler exponential decay model.

Our system solves MOT in cycles using the Divide and Conquer paradigm, breaking the problem down to the following tasks:**Prediction** prior to observing new data. Estimates are made on $k_{t}$ based on the past state $k_{t-1}$ and state transition $k_{t-1}\to k_{t}$ using the particle filters where particle motion is governed by our behavioral motion model, Section 5.1 and Section 5.2.**Association** of newly detected pedestrians (observations) $d_{i}$ to the predicted targets $k_{j}$. An assignment function $\Lambda \left(a_{i,j}\right)$ based on the observation likelihood maximizes the matching similarity in a combined 2D–3D hyperspace. The result of the association is a list of feasible tuples $\left\{\left(i,j\right)\right\}$, more details in Section 6.1.**Management** which finds an optimal policy for creation, update, deletion or merger of new targets to the tracker based on the association result L. Target updates are performed using the bootstrap particle filter formulation, details in Section 4.2, while creation, deletion, and merger of targets is performed based on our track confidence score χ, Section 6.3.

## 4. Proposed 2D–3D MOT Tracker

Tracking the position, velocity, and appearance of a single person from noisy image and LiDAR observations is not trivial. However, using the Markovian assumption, there exist methods with well-founded solutions to the problem which will be explained and extended in this section. We adopt the Bayesian tracking principle and apply the standard set of Bayesian tracking equations to a non-parametric state variable PDF. The numerical solution, i.e., the aximum a posteriori (MAP) solution of the pedestrian state is found by 2D and 3D particle filters. The novelty of our approach lies in the usage of behavioral motion models as state transition priors which make the filter capable of modeling pedestrian states in cases of unpredictable and ambiguous motion.

### 4.1. Bayesian Tracking

Each target in our system is a model of an independently tracked pedestrian. After optimally assigning new observations to the respective targets, we update the respective target model using Bayesian reasoning. We assume that the processes from which we seek to infer some information belong to the family of hidden Markov models (HMM) [37]. This model describes stochastic processes characterizing the evolution in discrete time of two sets of random variables: the *n*-dimensional state of a person at time *t* which is the state vector $k_{j,t}=\left\{r_{j},c_{j},▵r_{j},▵c_{j},w_{j},h_{j},app_{j},x_{j},y_{j},▵x_{j},▵y_{j}\right\}\in R^{n},$ and a respective observation vector, at time *t*: $d_{i,t}=\left\{r_{i},c_{i},w_{i},h_{i},▵r_{j},▵c_{j},app_{i},x_{i},y_{i},s_{i}\right\}\in R^{n}$. In a broader context, the problem of estimating the posterior probability $p\left(k_{t}\right)$ of the state vector $k_{t}$ at different times is formulated as the following sub-problems:(3)$$\left\{\begin{array}{cc} p\left(k_{t-n}\left|d_{1:t}\right.\right);\forall 1\geq n>t & \mathrm{smoothing},\\ p\left(k_{t}\left|d_{1:t-1}\right.\right) & \mathrm{prediction},\\ p\left(k_{t+n}\left|d_{1:t}\right.\right);\forall n\geq 0 & \mathrm{update}. \end{array}\right.$$

In our case, we are mainly interested in the latter two, where we sequentially estimate the current state of the tracked person and update the model with new measurements upon positive association. Unassociated tracks do not update and the final estimate is made only from the motion model. For each individual target, we formally write the probabilistic tracking problem as the following state space set of equations:(4)$$p\left(k_{t}\left|k_{t-1}\right.\right):k_{t}=f\left(k_{t-1},\xi_{t-1}\right),$$
(5)$$p\left(d_{t}\left|k_{t}\right.\right):d_{t}=h\left(k_{t},\eta_{t}\right),$$ where $\xi_{t-1}$ is the system noise and $\eta_{t}$ is the measurement noise. The system noise is a random variable which represents the inaccuracy of the state transition or prediction model $f\left(\cdot \right)$ and the measurement noise represents the error in the actual model position or state and the measurement, i.e., the sensor model $h\left(\cdot \right)$. Thus, the state to be estimated is a function of the previous state and a noise component, whereas the observed output is a function of the current state and a noise component. The goal of the tracking system is to estimate the optimal state $k_{t}$ subject to maximizing the belief in that state given past observations $d_{1:t}$. In Bayesian tracking, this is done by estimating the posterior density function $p\left(k_{t}\left|d_{t-1}\right.\right)$ using Equations (Equation 4) and (Equation 5).

During the prediction step, only the previous state estimate, $p\left(k_{t}\left|d_{t-1}\right.\right)$ is known and no observations are available yet. At this point, the past state is propagated into the current time by using the state transition probability $p\left(k_{t}\left|k_{t-1}\right.\right)$:(6)$$p\left(k_{t}\left|d_{t-1}\right.\right)=\int p\left(k_{t}\left|d_{1:t-1},k_{t-1}\right.\right)p\left(k_{t-1}\left|d_{1:t-1}\right.\right)dk_{t-1},$$ or, by dropping the observation from the state transition term, we get: (7)$$p\left(k_{t}\left|d_{t-1}\right.\right)=\int p\left(k_{t}\left|k_{t-1}\right.\right)p\left(k_{t-1}\left|d_{1:t-1}\right.\right)dk_{t-1}.$$

This equality is only true under the assumption that our tracklet is a first-order Markov process, i.e., the current state is independent of the previous observations if we know the previous state.

During the update step, a new observation $d_{i,t}$ becomes available and associated with the respective tracklet $k_{j,t}$ which we combine with the last state and observation pair (dropping the indices i,j) $k_{t},d_{t-1}$ using the Bayes’ rule:(8)$$p\left(k_{t}\left|d_{1:t}\right.\right)=\frac{p\left(d_{1:t}\left|k_{t}\right.\right)p\left(k_{t}\right)}{p\left(d_{1:t}\right)}.$$

Since we assumed that the tracklet is a first order Markov process, we can rewrite Equation (Equation 8) as:(9)$$p\left(k_{t}\left|d_{1:t}\right.\right)=\frac{p\left(d_{1:t-1}\left|k_{t}\right.\right)p\left(d_{t}\left|k_{t}\right.\right)p\left(k_{t}\right)}{p\left(d_{1:t-1}\right)p\left(d_{t}\left|d_{1:t-1}\right.\right)}=\frac{p\left(k_{t}\left|d_{1:t-1}\right.\right)p\left(d_{t}\left|k_{t}\right.\right)}{p\left(d_{t}\left|d_{1:t-1}\right.\right)}.$$

The first term in the numerator is the result of the prediction step (Equation 7) and the second term is the likelihood function from an observation model. It computes how likely the current observation $d_{t}$ is given the hypothesis state $k_{t}$. The denominator term $p\left(d_{t}\left|d_{1:t-1}\right.\right)$ can be expanded using the two known distributions:(10)$$p\left(d_{t}\left|d_{1:t-1}\right.\right)=\int p\left(d_{t}\left|k_{t}\right.\right)p\left(k_{t}\left|d_{1:t-1}\right.\right)dk_{t}.$$

It is worth mentioning that, if this PDF is normally distributed, and if both the motion model as defined by Equation (Equation 4) and the observation model as defined by Equation (Equation 5) are linear functions of their input, then the corresponding prior and likelihood distributions are inherently Gaussian too. In such case, where all distributions are Gaussian, and all transformations are linear, the integrals that are shown in Equations (Equation 7) and (Equation 10) can be computed analytically resulting in the linear Kalman filter. However, in our tracking problem, and given the update frequency of the sensors (10 Hz), these PDFs are far from Gaussian, and the pedestrian 3D motion model is definitely not linear. We offer an approximate solution to the posterior distribution by means of Monte Carlo sampling, while a high-level track management apparatus solves for the target to target interaction.

### 4.2. Particle Filters

Particle filters do not provide analytical solutions for Equation (Equation 9), but, instead, use Monte Carlo simulation to directly represent the posterior PDF as a weighted sum of *N* discrete samples (particles):(11)$$p\left(k_{t}\left|d_{1:t}\right.\right)\approx \sum_{i=1}^{N}w_{t}^{i}\delta \left(k_{t}-k_{t}^{i}\right),$$ where $k_{t}^{i}$ is a random sample from this distribution: $k_{t}^{i}∼p\left(k_{t}\left|d_{1:t}\right.\right),$
δ is the Dirac delta function and $w_{t}^{i}$ are sample weights, initially $w_{t}^{i}=\frac{1}{N}.$ In practice, drawing samples from the posterior is impossible because the posterior distribution is exactly what we are trying to estimate. On the other hand, for a given observation $d_{t}$, the likelihood $p\left(d_{t}\left|k_{t}^{i}\right.\right)$ can be computed relatively easy from the observation model defined by Equation (Equation 5). If a distribution cannot be sampled directly, but the likelihood of its samples can easily be evaluated, an approximation of this distribution can be obtained by means of importance sampling. Instead of sampling the posterior distribution, samples are drawn from any other reasonably chosen proposal distribution where the support of the samples must span over the posterior distribution. The sample weights $w_{t}^{i}$ from this proposal distribution are obtained by evaluating these samples using the likelihood function, such that the weighted set of samples approximates the true posterior distribution.

If we choose a proposal distribution $q\left(\cdot \right)$, using Bayes’ rule, we can re-write the posterior $p\left(k_{t}\left|d_{1:t}\right.\right)$ as:(12)$$p\left(k_{t}\left|d_{1:t}\right.\right)=\alpha p\left(d_{1:t}\left|k_{1:t}\right.\right)p\left(k_{t}\right),$$ where α is a normalization factor equal for all samples. The importance weight of each particle can be computed using the following equation:(13)$$w_{t}^{i}=\frac{p\left(d_{1:t}\left|k_{t}^{i}\right.\right)p\left(k_{t}^{i}\right)}{q\left(k_{t}^{i}\left|d_{1:t}\right.\right)},$$ which can be computed recursively [38] as:(14)$$w_{t}^{i}=w_{t-1}^{i}\frac{p\left(d_{t}\left|k_{t}^{i}\right.\right)p\left(k_{t}^{i}\left|k_{t-1}^{i}\right.\right)}{q\left(k_{t}^{i}\left|k_{1:t-1}^{i},d_{1:t}\right.\right)},$$ where the numerator is the product of the observation model $p\left(d_{t}\left|k_{t}^{i}\right.\right)$ and the motion model $p\left(k_{t}^{i}\left|k_{t-1}^{i}\right.\right)$ and the denominator is the proposal distribution. Particle filters that use recursive importance sampling to obtain a weighted, discrete approximation of the posterior distribution are called Sequential Importance Sampling (SIS) particle filters. The complete set of particles represents a probability mass function, such that the final state estimate $\hat{k}$ can be obtained by searching for the mode of this discrete distribution, or in some cases, if the posterior is believed to be unimodal and symmetric, by calculating the average:(15)$$\begin{array}{c} \hat{k}=E\left(k_{i}\right)=\mathrm{mode}\left(w_{i}k_{i}\right),\\ \hat{k}=E\left(k_{i}\right)=\frac{\sum_{i}^{N}w_{i}k_{i}}{\sum_{i}^{N}w_{i}}. \end{array}$$

We employ the Kernel Density Estimation technique (KDE) to find the approximate mode of the posterior. This process can be parallelized and trades memory to achieve fast execution. We project the coordinates of each particle $k_{i}$ onto a small and finite 2D accumulator grid using a pyramid kernel. Then, we find the position of the peak by fitting a parabola around the maximum in the accumulator grid. Thus, our estimates $\hat{k}$ on the image and on the ground plane $E\left(k_{i}^{\mathrm{image}}\right)$ and $E\left(k_{i}^{\mathrm{ground}}\right)$ are at the peak of the respective parabolas. This means that we assume the state variable posteriors to be locally Gaussian, but, in theory, can have as many modes as the number of particles.

At each time step, we check if an observation has been assigned to each track. For tracks that have been associated with observations, we re-weigh each particle based on the observation likelihood function. Particles that reside in regions of space that is more likely contribute more to the posterior than other particles. After each positive association, we produce new particles by multiplying ones with higher weights and removing the others. In order to tackle the problem of particle depletion and impoverishment, we employ Sequential Importance Resampling (SIR) of the particles using the multinomial resampling technique. In multinomial resampling, the set of *N* new particles is sampled with replacement from the set of *N* old particles. The probability that a particle *i* is picked at each sampling round is defined by its weight $w^{i}$. After resampling, all weights are set to $w^{i}=\frac{1}{N}$ such that the new sample set represents the same distribution as the original sample set before resampling. In general, we want to re-sample as little as possible, while still being able to model the mode of the posterior density accurately. Hence, a proposal distribution that closely matches the posterior distribution is extremely important as it greatly reduces the need for resampling.

The optimal proposal distribution has been shown to be the one that minimizes the variance of the importance weights [38]. However, in practice, it is usually impossible to calculate this optimal distribution. Most current implementations, such as the well known CONDENSATION algorithm [39], simply use the state transition prior $p\left(k_{t}^{i}\left|k_{t-1}^{i}\right.\right)$, i.e., the target motion model, as the proposal distribution. Particle filter implementations that employ this strategy are called bootstrap filters and are the most widely used particle filter variant. They ignore the fact that the proposal distribution is conditioned on the latest observation. Instead, they assume that the current state is only a function of the previous state and is independent from the latest observation. In other words, the posterior is assumed to change smoothly over time thereby closely resembling the transition prior at each time step *t*. If we then plug the state transition prior into (Equation 14), we simplify the weight computation into:$$w_{t}^{i}=w_{t-1}^{i}\frac{p\left(d_{t}\left|k_{t}^{i}\right.\right)p\left(k_{t}^{i}\left|k_{t-1}^{i}\right.\right)}{p\left(k_{t}^{i}\left|k_{t-1}^{i}\right.\right)}=w_{t-1}^{i}p\left(d_{t}\left|k_{t}^{i}\right.\right),$$ which cancels out the proposal distribution in the bootstrap filter equation. The latest observations are then used to assign a likelihood (weight) to each particle. In the case that an observation is not available, the particles are moved according to $p\left(k_{t}^{i}\left|k_{t-1}^{i}\right.\right)$ and their weights remain the same. This way, we greatly simplify the computation of the particle’s weight updates and subsequent re-sampling since we are able to re-sample the particles regardless of whether new data is available or not. We note that our behavioral 3D motion model, discussed in Section 5, that closely resembles pedestrian motion is applied independently to every particle and allows the particles to spread in a pattern which closely matches the expected posterior. This means that each particle is a stochastic process which introduces noise and disperses some particles to the tails of the posterior. We re-sample the particles only when the Effective Sample Size (ESS) metric becomes too low: ${\left(\sum w^{i}\right)}^{-1}<0.5N.$

## 5. Motion and Observation Models

### 5.1. Ground Plane Motion Model

In our particle filter, we integrate an accurate motion model based on short-term pedestrian behavior. Our model is learned from an annotated database containing short- to medium-term tracked pedestrians in 2D images, namely the KITTI tracking database [1]. From these annotations, we are able to extract 3D trajectories by matching the calibrated point cloud to the pedestrian position in the image, Figure 4 left. Many approaches in the literature assume that the pedestrians move with constant velocity and spawn particles with a uniform starting velocity. In a very broad context, this assumption is true, however, in the traffic scenarios of the KITTI dataset, we measured a behavior which is highly non-uniform and far from constant.

**Initialization:** by experimental fitting of various reasonable functions to the measured velocity data, we found that the probability distribution for walking at a certain pace $∥v∥$ closely follows a truncated bi-modal Gaussian function, Figure 4 middle, with a small peak around 0 km/h and a stronger peak at 5 km/h:(16)$$p\left(∥v∥\right)=\left\{\begin{array}{cc} \sum_{i}a_{i}N\left(∥v∥;\mu_{i},\sigma_{i}\right) & \mathrm{if}∥v∥\in \left[0,10\right],\\ 0 & \mathrm{otherwise}, \end{array}\right.$$ with $a_{1}=0.176,\mu_{1}=0.838,\sigma_{1}=1.293,$$a_{2}=0.823,\mu_{2}=5.125,\sigma_{2}=1.024$ and $N\left(x;\mu ,\sigma \right)$ is:(17)$$N\left(x;\mu ,\sigma \right)=\frac{1}{\sqrt{2\pi \sigma^{2}}}exp\left(-\frac{{\left(x-\mu \right)}^{2}}{2\sigma^{2}}\right).$$

This indicates that most of the annotated pedestrians are either standing still or walking at a normal pace around 5 km/h. We exploit this walking pace prior in the process of particle initialization, where particles in our tracker will start with velocity magnitudes drawn from this prior distribution.

**Longitudinal acceleration:** pedestrians almost never walk at a constant pace due to external factors such as ground conditions, proximity to obstacles or other traffic. Additionally, high level stimuli cause a person to frequently speed up or slow down. We include this behavior in our motion model as a probability density function for the change in velocity magnitude. Within the KITTI dataset, this probability distribution for acceleration is independent from the velocity magnitude and follows a Gaussian function:(18)$$p\left(▵v\right)=N\left(▵v;\mu_{▵v},\sigma_{▵v}\right),$$ with $\mu_{▵v}=0.011,\sigma_{▵v}=0.809$. During each prediction step, $p\left(k_{t}^{i}\left|k_{t-1}^{i}\right.\right)$, we move each of our particles according to this motion model by changing the walking pace with values drawn from (Equation 18).

**Lateral velocity:** external stimuli can affect motion also in the lateral direction. Although there are models for human motion based on sentiment analysis, most of it has limited application in a wide traffic context. Without a comprehensive dataset, accurate long-term modeling of human behavior can only stay theoretical and is difficult to evaluate. Therefore, we focus on the short term probability to change the direction of walking (Yaw Rate) which we found to be conditioned on the longitudinal velocity, Figure 4 right. We noticed that the faster a person is moving, the less likely that person is to change the direction of travel, thus the probability distribution for the change of travel direction ▵θ is:(19)$$p\left(▵\theta \left|∥v∥\right.\right)=N\left(▵\theta ,∥v∥;0,\sigma_{▵\theta }\right),$$ where the standard deviation $\sigma_{▵\theta }$ is a function on the longitudinal velocity and follows a bi-modal Gaussian distribution:(20)$$\sigma_{▵\theta }\left(∥v∥\right)=\sum_{i}a_{i}N\left(∥v∥;\mu_{i},\sigma_{i}\right),$$ with parameters: $a_{1}=105.4,\mu_{1}=-20.73,\sigma_{1}=11.81,a_{2}=48.14,\mu_{2}=0.58,\sigma_{2}=0.95$. If we observe the data points on the right plot in Figure 4, we can clearly see this property, e.g., when the pedestrian velocity is low or close to zero, the dispersion of the measured yaw rates is extreme and almost uniform across the range $\left[-\pi ,\pi \right]$. This means that a still person can start to move in virtually any direction with an equal likelihood. Likewise, when the person is walking at a faster pace, the dispersion of the yaw rate is closer to 0, which means that, as the walking pace increases, we see a clear reduction in the yaw rate extent. This notion is somewhat intuitive because of the momentum that the human body accumulates with higher velocity. A simulation of 20 particles following the above-mentioned motion models is given in Figure 2.

**Particle updates:** prior to observing new data $d_{t},$ we move each particle by first changing their direction of motion. Our motion model $p\left(k_{t}^{i}\left|k_{t-1}^{i}\right.\right)$ on the ground plane consists of sampling a new longitudinal and lateral velocity based on the previous values and the models in (Equation 18)–(Equation 20) and applying this new motion to the previous coordinates of each particle. Formally, for each particle *i*, $k_{t}^{i}$ at time *t* with previous position $x_{t-1}^{i},y_{t-1}^{i}$ and velocity $▵x_{t-1}^{i},▵y_{t-1}^{i},$ we do the behavioral motion update in polar coordinates:(21)$$∥v_{t}^{i}∥=∥v_{t-1}^{i}∥+v^{\prime };v^{\prime }∼N\left(\mu_{▵v},\sigma_{▵v}\right),$$
(22)$$\theta_{t}^{i}=\theta_{t-1}^{i}+\theta^{\prime };\theta^{\prime }∼N\left(▵\theta ,∥v∥;0,\sigma_{▵\theta }\right),$$ where the change in longitudinal and lateral velocities $v^{\prime },\theta^{\prime }$ are random samples from the respective distributions. New particle positions $x_{t}^{i},y_{t}^{i}$ are then computed by transforming this motion back to Cartesian coordinates:(23)$$\begin{array}{c} x_{t}^{i}=x_{t-1}^{i}+∥v_{t}^{i}∥\cos\left(\theta_{t}^{i}\right),\\ y_{t}^{i}=y_{t-1}^{i}+∥v_{t}^{i}∥\sin\left(\theta_{t}^{i}\right), \end{array}$$ and particle velocities are updated accordingly:(24)$$\begin{array}{c} ▵x_{t}^{i}=∥v_{t}^{i}∥\cos\left(\theta_{t}^{i}\right),\\ ▵y_{t}^{i}=∥v_{t}^{i}∥\sin\left(\theta_{t}^{i}\right). \end{array}$$

### 5.2. Image Plane Motion Model

Applying behavioral analysis for bounding box motion in images from a moving vehicle is very difficult since positions and velocities depend on the relative position and motion between the camera and the objects. Additionally, due to the perspective projection model, even objects with constant ground plane velocity result in nonlinear image plane velocity. Nevertheless, translating the motion model of the ground plane to the image plane is possible, but not desirable due to reasons discussed in Section 3. To that end, our image plane particles move independently from the ones on the ground plane. The particle filter uses an adaptive acceleration motion model to update each particle position and velocity.

**Initialization:** since it is difficult to have an informative prior for the general position of a pedestrian in the image plane, we initialize particle positions by drawing samples from a 2D Gaussian distribution around the first observation location $\left(r_{i},c_{i}\right)$:(25)$$\begin{array}{c} \begin{array}{c} p\left(r,c\right)=N\left(r,c;\mu_{r,c},\sum_{r,c}\right), \end{array}\\ \mu_{r,c}=\left(r_{i},c_{i}\right),\sum_{r,c}=\left[\begin{array}{cc} 32 & 0\\ 0 & 32 \end{array}\right], \end{array}$$ where the covariance matrix Σ is expressed in pixel values. For initialization of image plane particle velocities, we resort to observation data, namely the optical flow field values $\left(\Delta r^{i},\Delta c^{i}\right)=\left(\Delta r_{OF}^{i},\Delta c_{OF}^{i}\right)$ at each particle position.

**Longitudinal velocity:** during estimation, we apply the following motion model to each particle position and velocity (for simplicity we re-use the symbol *v* for the image plane velocity):(26)$$\begin{array}{c} \begin{array}{c} ∥v_{t}^{i}∥=∥v_{t-1}^{i}∥+v^{\prime }\\ v^{\prime }∼N\left(0,\sigma_{▵v}\right), \end{array}\\ \sigma_{▵v}\propto \left(v_{t-1},w_{t-1}\right), \end{array}$$ where the image plane velocity update span $\sigma_{▵v}$ increases proportional to the image plane velocity of the tracklet *v* and the size of the image bounding box *w*. The motivation for using this formulation is easy to see, since the particles must accommodate the case where a target is getting closer to the camera, i.e., *w* increases. In this case, the same amount of ground plane motion can result in larger image plane displacement. On the other hand, targets that are far away and have small bounding boxes will be modeled by particles that change their velocity more slowly.

**Lateral velocity:** the change of direction of each particle in the image plane is modeled using the following angular acceleration model (again using the same notation θ for simplicity) where: (27)$$\begin{array}{c} \theta_{t}^{i}=\theta_{t-1}^{i}+\theta^{\prime },\\ \theta^{\prime }∼N\left(▵\theta ;0,\sigma_{▵\theta }\right), \end{array}$$ meaning each particle’s direction of travel can change according to a normal distribution with standard deviation of $\sigma_{▵\theta }=0.4$ radians. This allows our particles to explore the image position state space efficiently prior to observing new data.

**Particle updates:** at each time step, we adaptively introduce information from the optical flow field into the motion model by adding the *x* and *y* motion vectors to the values computed in (Equation 26) and (Equation 27). We formulate a coefficient of Optical Flow trust, based on the current EGO vehicle velocity, where the influence of the optical flow in the particle motion increases relative to the EGO velocity. This is controlled by the innovation coefficient γ:(28)$$\gamma =1-\exp\left(-\frac{{∥v_{EGO,t}∥}^{2}}{\sigma_{EGO}^{2}}\right),$$ where the velocity $∥v_{EGO,t}∥$ is expressed in $kmh^{-1}$ and $\sigma_{EGO}$ is chosen such that, when the velocity is low, then γ is close to 0, and, when the velocity is high, then γ approaches 1. Our image plane particle velocity update is:(29)$$\begin{array}{c} \Delta r_{t}^{i}=\gamma \Delta r_{OF}^{i}+\left(1-\gamma \right)∥v_{t}^{i}∥\cos\left(\theta_{t}^{i}\right),\\ \Delta c_{t}^{i}=\gamma \Delta c_{OF}^{i}+\left(1-\gamma \right)∥v_{t}^{i}∥\cos\left(\theta_{t}^{i}\right), \end{array}$$ where $\Delta r_{OF}^{i}$ and $\Delta c_{OF}^{i}$ are the *x* and *y* components of the optical field values at the respective particle positions. This means that, when the vehicle is static, we don’t use optical flow vectors to update our particle motion, when the vehicle is moving fast we only rely on the optical flow vectors and, for the cases in between, we use a combination of both the optical flow and the particle random motion. The new particle positions are computed in Cartesian image coordinates—for brevity, refer to (Equation 23).

### 5.3. Observation Likelihood Model

The observation likelihood measures how well the current configuration of target states $\left\{k_{1},k_{2},\ldots ,k_{m}\right\}$ explains the observation data $\left\{d_{1},d_{2},\ldots ,d_{n}\right\}$. In our tracker, we use the observation likelihood as a critical similarity measure for solving the data association problem, but also for updating the track confidence score, Section 6.

Since we break down the MOT problem into tracking individual targets separately, our observation likelihood is simplified and only explains how well each specific detection $d_{i}$ matches each specific person $k_{j}$. Our method projects the MAP solution, ${\hat{k}}_{i}$, of the posterior $p\left(k_{t}\left|d_{1:t-1}\right.\right)$ into one of the input space using $f_{k}\left({\hat{k}}_{i}\right)$, and then computes the observation likelihood function $p\left(d_{j,t}\left|f_{k}\left({\hat{k}}_{i}\right)\right.\right)$ given the input data $d_{j}$. Thorough explanation of each projection function $f_{k}\left({\hat{k}}_{i}\right)$ is outside of the scope of this paper. We hereby follow the approach of [33] and model observational model using log-likelihood functions for the different input spaces. The detection modalities are the following: ground plane position (obtained from LiDAR segmentation), image plane position and size size (obtained from the detector of [40]), appearance, obtained from Hue-Saturation-Value (HSV) histograms, and image plane velocity (obtained from optical flow [35]). Each constituent log-likelihood function describes the observational error in one of the input spaces. While in the literature the observation model is often given ad hoc using approximations with Gaussian functions, in this study, we learned the model parameters by performing a thorough examination of the object detector behavior against ground truth data.

At time *t*, the likelihood that detection $d_{t}$ originates from tracklet $k_{i}$ is:(30)$$p\left(d_{t}\left|f_{k}\left({\hat{k}}_{i}\right)\right.\right)=\exp\left(\sum_{j}-l_{j}\left(d_{t}\left|f_{k}\left({\hat{k}}_{i}\right)\right.\right)\right),$$ where each function $l_{j}()$ computes the likelihood that our observation originates from the target in the respective modalities:

**Ground plane center:** if a person modeled by track $k_{i}$ exists at ground plane location $\left(x_{i},y_{i}\right),$ then positions in close proximity are likely to be parts of a Point Cloud segment detected by the LiDAR. The observation likelihood $l_{GP}()$ is computed as ground plane Euclidean distance between the detection $d_{j}$ and the ground plane MAP solution ${\hat{k}}_{i}^{ground}$:(31)$$l_{GP}\left(d_{t}\left|f_{1}\left({\hat{k}}_{i}\right)\right.\right)=l_{GP}\left(d_{t}^{ground}\left|{\hat{k}}_{i}^{ground}\right.\right)=\frac{∥d_{t}^{ground},{\hat{k}}_{i}^{ground}∥}{\sigma_{1}^{2}}.$$

We use the normalizing constant $\sigma_{1}$ which we learned offline by fitting LiDAR segments positions to ground truth 3D object bounding boxes in the KITTI object detection dataset.

**Image center and size:** similarly, if a person $k_{i}$ exists and their projection on the image plane, $f_{k}\left({\hat{k}}_{i}\right)$, is defined by a bounding box center $\left(r_{i},c_{i}\right),$ then the observation likelihood $l_{IPcenter}()$ that a detection $d_{t}^{image}$ at $\left(r_{j},c_{j}\right)$ originates from this person is the Euclidean distance between the detection BB and the BB of the MAP solution ${\hat{k}}_{i}$ on the image plane:(32)$$l_{IPcenter}\left(d_{t}\left|f_{2}\left({\hat{k}}_{i}\right)\right.\right)=l_{IPcenter}\left(d_{t}^{image}\left|{\hat{k}}_{i}^{image}\right.\right)=\frac{∥d_{t}^{image},{\hat{k}}_{i}^{image}∥}{\sigma_{2}^{2}},$$ where the constant $\sigma_{2}$ controls the extent of the likelihood function and is empirically computed for the respective object detector based on available ground truth data in the KITTI 2D object detection dataset. The second image plane cue is the bounding box size, i.e., the likelihood that a person $k_{i}$ would be detected as a bounding box $d_{t}$ with a certain diagonal in pixel values, $\delta \left(k_{i}\right)=\mathrm{diag}\left(k_{i}\right)=\sqrt{w_{i}^{2}+h_{i}^{2}}$. Formally, $l_{IPdiag}()$ computes the relative diagonal difference of the detection and the projected MAP estimate:(33)$$l_{IPdiag}\left(d_{t}\left|f_{3}\left({\hat{k}}_{i}\right)\right.\right)=\sigma_{3}^{-2}{\left|\frac{\delta \left(d_{t}\right)-\delta \left({\hat{k}}_{i}\right)}{min\left(\delta \left(d_{t}\right),\delta \left({\hat{k}}_{i}\right)\right)}\right|}^{2}.$$

We note that, in the literature, the image plane observation likelihood is often modeled using the Jaccard Index i.e., the so-called Intersection over Union (IoU) metric. However, we found that using IoU produces 0 value for BBs that are not intersecting which is a brittle approach since it creates an unnatural discontinuity in the observation likelihood. A strong argument case is when using a low-precision object detector where it is possible that none of the detection BBs intersect any of the targets in which case the IoU is not discriminative.

**Image appearance:** If a person $k_{i}$ exists in the image plane with a certain appearance, then the pixels within the bounding box are likely to be observed even when parts of the person are occluded or the person is not facing the camera straight. The image patch under the detection bounding box $d_{j}:\left(r_{j},c_{j},w_{j},h_{j}\right)$ is compared to the image patch under the tracklet MAP by computing the appearance of the solution ${\hat{k}}_{i}:\;f_{k}\left({\hat{k}}_{i}\right)$. This task is a typical person re-identification problem where a specific person is queried in a database of known identities. Several studies [41,42] have shown that HSV and YCbCr features exhibit superior person re-identification performances over other features, i.e., these features are highly sensitive to the unique visual appearance of each person. Thus, the log-likelihood appearance value for each detection $d_{t}$ is computed by comparing the normalized 48 bin HSV color histogram to the HSV histogram of the target. In order to capture the variability in upper and lower body clothing, and to some degree make it robust to partial occlusions, we compute and compare two histograms for the regions $r^{top}\in \left[r_{t}-\frac{h_{t}}{2},r_{t}\right)$ and $r^{bottom}\in \left[r_{t},r_{t}+\frac{h_{t}}{2}\right)$. The appearance observation log-likelihood $l_{app}()$ is then the Kullback–Leibler divergence (KLD) between the histograms of the detection and histograms of the target, for simplicity *P* and *Q*:(34)$$D_{KL}\left(P∥Q\right)=\sum_{i}^{nBins}P\left(i\right)log\left(\frac{P\left(i\right)}{Q\left(i\right)}\right).$$

This formulation captures the relative entropy of *P* with respect to *Q*, i.e., the information difference for re-identification of our newly detected person with another already tracked person. In other words, it is the amount of information lost when a detection is used to approximate a stored tracklet. In applications, *P* typically represents the “true” distribution of data, observations, or a precisely calculated theoretical distribution, while *Q* typically represents a model or approximation of *P*. In order to find a distribution *Q* that is closest to *P*, we can minimize KL divergence and compute an information projection [43]. Since the KLD function is not symmetric, $D_{KL}\left(P∥Q\right)\neq D_{KL}\left(Q∥P\right)$, we compute the symmetric form for each color channel as:(35)$$D_{KL}\left(d,k;HSV\right)=\sum_{i:\left\{H,S,V\right\}}\left[D_{KL}\left(i_{d}∥i_{k}\right)+D_{KL}\left(i_{k}∥i_{d}\right)\right],$$ which plugs into our appearance likelihood:(36)$$l_{app}\left(d_{t}\left|f_{4}\left({\hat{k}}_{i}\right)\right.\right)=\frac{{∥D_{KL}\left(k_{i},d_{t}\right)∥}^{2}}{\sigma_{4}^{2}}.$$

The choice of the normalizing constant $\sigma_{4}$ is not trivial since it should balance the range of appearance log-likelihood values with respect to the other terms in Equation (Equation 30). For the sake of brevity, in this work, we empirically chose a value that gives equal importance of the appearance error relative to the other measurement errors.

**Image velocity:** the likelihood of observing a person $k_{i}$ moving with image plane velocity $k_{i}:\left(\Delta r_{i},\Delta c_{i}\right)$ is given by the difference of their velocity vector and the optical flow field under the observation bounding box. For this feature, we first need to compute the motion vectors from the video sequence or to sample the optical flow field and compute the Euclidean distance to the motion vectors of the target:(37)$$l_{vel}\left(d_{t}\left|f_{5}\left({\hat{k}}_{i}\right)\right.\right)=-\frac{{∥\Delta r_{i}-\Delta u_{t},\Delta c_{i}-\Delta v_{t}∥}_{2}^{2}}{w_{t}\sigma_{5}^{2}},$$ where the spatial motion vector $\left(\Delta u,\Delta v\right)$ explains how the brightness of the current frame matches the brightness of a frame in the past time instance t−1. Practically, we use the average optical flow motion vector within the upper-central (human torso) area of a bounding box. The constant $\sigma_{5}$ here controls the expected “spreading” of the detection’s vector direction and orientation relative to the target and is chosen empirically to give equal balance of this term in the likelihood function Equation (Equation 30).

## 6. Data Association and Track Management

Performance of multi-target particle filter tracking suffers from track interactions and multiple observations in close proximity. Therefore, it is very important to handle such situations with care in order to reduce false track generation, track switches and/or false track merger. We separate the data association problem from the tracking such that each observation can update at most one target and each target can be updated by, at most, one observation. From a theoretical point of view, the separation of association from the tracking can be seen as applying a spatial gating function to each tracklet so that only observations that fall within the gating range can update the respective target. This technique is very common in MOT literature as it is often applied to large problems in order to obtain faster operation. After assignment, each track is modeled as a separate Markov Decision Process (MDP) and is subjected to a track management system which handles high-level actions such as adding a new track, deletion of an old track, merging of track or re-identification. Designing our MOT system as a separate assignment, tracking and interaction processes have multiple benefits over a monolithic tracker.

Firstly, we don’t need to update all particles with every observation. We form state predictions $p\left(k_{t}\left|d_{1:t-1}\right.\right)$ from the previous state $p\left(k_{t-1}\left|d_{1:t-1}\right.\right)$ and the state transition functions $p\left(k_{t}\left|k_{t-1}\right.\right)$ by moving each particle according to the motion models in Section 5.1 and Section 5.2. Then, we find the optimal association of new observations $d_{j}$ and the Maximum a posteriori (MAP) estimates of each target ${\hat{k}}_{t}$, by computing the observation likelihood function $p\left(d_{j,t}\left|{\hat{k}}_{i,t}\right.\right)$ from Section 5.3.

Second, more complex track interaction schemes can be devised on the basis of track status and history. A track management apparatus decides which of these actions can be taken by finding an optimal tracking policy π. Lastly, our system is modular meaning that sub-problems can be solved using different algorithms trading computing power for memory and vice versa.

### 6.1. Optimal Data to Target Association

The fundamental problem of assigning observations to targets can be analyzed from a combinatorial optimization perspective. Formally, we want to assign all observations $d_{j}$ to existing tracklets using their MAP estimates ${\hat{k}}_{i}$. As discussed in Section 4.2, each individual tracker is a bootstrap PF where the proposal function is the state transition prior (which does not depend on the latest observation $d_{j}$). This formulation allows us to make an unbiased matching of observations and state estimates. At the time of matching, it is possible to have more targets than observations or vice versa. Additionally, as seen in Figure 5, some of the associations are unfeasible due to physical limitations. Prior to solving the association problem, we impose gating to restrict impossible combinations. Only observations $d_{j}$ with detection score $s_{i}>0.2$ are considered for an association. In addition, an association is considered feasible only if the computed observation likelihood is above a threshold $p\left(d_{j}\left|{\hat{k}}_{i}\right.\right)>0.5$. With these rules in mind, we optimize the measurement to target association as follows. The assignment problem looks for the optimal set of combinations of targets (agents) to detections (tasks). Any agent can be assigned to perform any “realistic” task with a corresponding weight or cost of doing the respective task. All tasks need to be performed by assigning exactly one agent to each task and exactly one task to each agent in such a way that the total cost of the assignment is minimal.

We formulate this problem as a complete bipartite graph $K_{m,n}$, Figure 6, with bipartition $\left\{X,Y\right\},$ where $\left|X\right|=m,\left|Y\right|=n$, and each vertex of *X* is adjacent to every vertex of *Y*. At time t=T, the first set of vertices, *X*, is composed of the MAP target estimates $\left\{{\hat{k}}_{1},{\hat{k}}_{2},\ldots ,{\hat{k}}_{m}\right\}$ and the second set of vertices, *Y*, represents our observations, i.e., the pedestrian detections $\left\{d_{1},d_{2},\ldots ,d_{n}\right\}$ where each detection $d_{i}$ consists of the image plane and ground plane positions, sizes and detection scores $d_{i}:\left(r_{i},c_{i},w_{i},h_{i},s_{i},x_{i},y_{i}\right)$. If *N* is a network from $K_{m,n}$ where each edge is an assignment $a_{i,,j}$ with a weight $\Lambda \left(a_{i,,j}\right)$ equal to the observation likelihood function computed in (Equation 30):(38)$$\Lambda \left(a_{i,,j}\right)=p\left(a_{i,,j}\right)=p\left(d_{j}\left|{\hat{k}}_{i}\right.\right).\;$$

We want to find the maximum weight in the network *N*. Following the Kőnig’s theorem [44], which states that there is an equivalence between the maximum matching problem and the minimum vertex cover problem in bipartite graphs, the algorithm of Kuhn (Hungarian algorithm, Munkres algorithm) [45] can be used to find the list L with maximum weights in our network in $O\left(n^{3}\right)$:(39)$$\begin{array}{c} L=\mathrm{argmax}\\ \;\;\;\;\;\;\;\;\;\;\;\;\;\;\left\{\left(i,j\right)\right\} \end{array}\begin{array}{c} p\left(d_{j}\left|{\hat{k}}_{i}\right.\right). \end{array}$$

The list of association tuples $L:\left\{\left(i,j\right)\right\}$ contains all physically feasible associations of observations and targets. This means that some of the targets might not be included in this list as well as some of the observations. In our practical application, the Hungarian algorithm is able to quickly find an optimal assignment solution since the number of detections and observations are relatively small, $n\cdot m\in \left(10^{0},10^{3}\right)$. Due to the nature of the definition of our cost function (Equation 30), the solution is optimal in the multi-modal 2D–3D space. By using the sum of log-likelihood functions, we allow our solution to be robust to small variations in position and appearance; however, greater discrepancies in one or more constituent metrics will give a low matching cost. Having found the optimal measurement to target association tuples, we apply our track management apparatus to each target where we also allow for track-to-track interactions to occur.

### 6.2. Track Confidence Score

Critical to the operation of our MOT is having a reliable measure of the quality of each track. This is necessary in order to distinguish between actual pedestrians, spurious false positives and consistent clutter and only output tracks which the tracker is confident are actual pedestrians. Broadly speaking, there are several factors that can influence the confidence of a track. Firstly, the number of positive observation associations over time, then the number of missing observations, the probability that the observations actually come from the tracked pedestrian and the probability that the observations are background. Finally, track confidence is also affected by the quality of association of the observations to the track.

Formally, we follow the track confidence score, $S_{t}$, formulation of [10,36] by computing the measurement log likelihood ratio (LLR) over the lifespan of each track $k_{i}$. As discussed in [10], the track score consists of a motion and appearance term, but since the motion score is difficult to compute analytically in the case of a particle filter tracker, in this paper, we only use the appearance score as our track confidence score. The appearance LLR measures the likelihood that observations $d_{1:t}$ associated with the track $k_{i}$ come from the actual pedestrian $\left(j_{1:t}\subseteq k_{i}\right),$ which is the target hypothesis, versus the background $\left(j_{1:t}\subseteq \emptyset \right)$, which is the null hypothesis:(40)$$S_{i}^{t}=\ln\frac{p\left(d_{j_{1:t}}\left|j_{1:t}\subseteq k_{i}\right.\right)}{p\left(d_{j_{1:t}}\left|j_{1:t}\subseteq \emptyset \right.\right)},$$ with the simplified notation $j_{1},j_{2},\ldots ,j_{t}$ for the sequence of observations. Given the Markovian assumption and Bayesian tracking formulation in Section 4.1, the LLR can be factored as:(41)$$\ln\frac{p\left(d_{j_{1:t}}\left|j_{1:t}\subseteq k_{i}\right.\right)}{p\left(d_{j_{1:t}}\left|j_{1:t}\subseteq \emptyset \right.\right)}=\ln\frac{\prod_{t}p\left(j_{t}\subseteq k_{i}\left|j_{1:t-1}\subseteq k_{i},d_{j_{1:t}}\right.\right)}{\prod_{t}p\left(j_{t}\subseteq \emptyset ,d_{j_{1:t}}\right)},$$ where $j_{t}\subseteq k_{i}$ is conditionally independent of any future measurements $d_{j}$ and the null hypotheses $j_{t}\subseteq \emptyset$ are also independent from the current measurement $d_{j_{t}}$. Simmilar to [10], we model each term in the numerator of Equation (Equation 41), i.e., the posterior, as the probability that the measurement $d_{j}$ is in track $k_{i}$ which we approximate with the the observation to target association score, Equation (Equation 38):(42)$$p\left(j_{t}\subseteq k_{i}\left|j_{1:t-1}\subseteq k_{i},d_{j_{1:t}}\right.\right)=\frac{e^{\Lambda \left(a_{i,,j}\right)}}{e^{\Lambda \left(a_{i,,j}\right)}+e^{-\Lambda \left(a_{i,,j}\right)}},$$ and we use a constant probability *c* for the null hypothesis posterior:(43)$$p\left(j_{t}\subseteq \emptyset ,d_{j_{1:t}}\right)=C.$$

In this formulation, tracks which originate from true pedestrians will likely have positive track confidence scores $S_{i}>0$ while tracks from false positives will have negative score S<0. It is easy to see that the track confidence score $S_{t}$ can be computed recursively, at each time step *t*, from the previous score $S_{t-1}$. Following [36], we have:(44)$$S_{t}=S_{t-1}+\Delta S_{t},$$
(45)$$\Delta S_{t}=\left\{\begin{array}{cc} \ln\frac{1-P_{D}}{1-P_{FA}}\approx \ln\left(1-P_{D}\right), & \mathrm{if}j_{t}\subseteq \emptyset ,\\ -\ln\left(1+\exp\left(-2\Lambda \left(a_{i,,j}\right)\right)\right)-\ln C, & \mathrm{otherwise}, \end{array}\right.$$ where $P_{D}$ and $P_{FA}$ are the prior probabilities of detection and false alarm. From this formula, it is clear that the track confidence score $S_{t}$ is unbounded, which can potentially lead to unwanted system behavior. For example, since our vehicle is moving and objects come in and out of view rather quickly, it is desirable that we decrease our confidence once the target gets out of the frame. At the same time, a target that has been lost for a longer period will have a very low confidence score, but if it suddenly re-appears, it will take a long time for the score to recover. Therefore, as discussed in the following sub-section, it is especially difficult to make the correct track management steps based directly on the track score in Equation (Equation 44). In order to solve the problems in such situations, we propose putting practical bounds on the track score, $\left|S_{t}\right|\leq B,$ and recover the probability of a true target, as in [46], using the sigmoid of the bounded track score $\chi_{t}=\mathrm{sigmoid}\left(S_{t}\right)$:(46)$$\chi_{t}=\frac{e^{S_{t}}}{1+e^{S_{t}}}.$$

The probability of a true target χ then has a probabilistic interpretation that is easy to integrate within a track management framework such as the Markov Decision Process.

### 6.3. Track Management

The proposed track manager handles the moves of individual targets, such as $\left\{add,update,remove,merge,reidentify\right\}$. Our technique is inspired by the state-of-the-art track management system proposed by Xiang et al. [47] which handles track jumps with priors that are learned offline. This method is currently one of the top performing published pedestrian trackers across various tracking benchmarks. Our track management, on the contrary, uses a set of novel, online, actions and reward functions based on the track confidence score and track to track interaction.

Each target $k_{i,1:t}$ is a separate Markov Decision Process (MDP) defined by the sets of states S, actions A and transition and reward functions $\left\{S,A,T,R\right\}$. States s∈S determine the logical status of the target such as being a inactive, active, confident or lost, green circles on Figure 7. An action a∈A can be performed on a target to create a state transition T∈T defined as T:S×A↦S, orange circles on Figure 7, with a probability distribution $P_{A}$ over states to which the process will transition when using action A. $P_{A,S}$ denotes the probability for transitioning to state S by taking action A. The reward function R∈R is defined as $R\left(T\left(S\times A\right)\right)\mapsto R$ computes the reward, or rather the cost, of executing the respective transition function comprised of action A and state S. Contrary to [47], in our model, we assume uniform $P_{A,S_{i}}$ over all possible outcomes S. Therefore, moves (jumps) will be governed by the quality of association and temporal stability. To that end, we exploit the temporal stability property form of our track confidence variable χ. This confidence variable can have values ranging between 0 and 1 and connects the association quality $p\left(d_{j}\left|{\hat{k}}_{i}\right.\right)$ with the temporal dimension, i.e., the number of times an association has been made as well as the probability for missclassification. It forms the backbone of our MDP by defining the track status $S_{i}$. At each time step, we update $\chi_{i}$ and based on the reward function *R* an action A is taken by target $k_{i}$. Every target in our system can have one of the following four statuses S:

**Inactive: $S_{k}^{inactive}$** defines the status of an unobserved, potential target with unknown state or a track for which we’ve lost our confidence and we are no longer tracking.

**Active: $S_{k}^{active}$** defines the status of a target which we are actively tracking, i.e., a target for which we make predictions and updates based on available observations.

**Confident: $S_{k}^{confident}$** is a special case of the active state. Confident tracks represent targets whose state variables are known with a high degree of confidence.

**Lost: $S_{k}^{lost}$** is the status of a target which becomes unobserved after being actively tracked. Such targets might re-appear in the camera frame after a short period of time and get re-identified by the tracker.

The goal of the track management system is to find a policy π which maximizes the reward by taking appropriate actions given the probabilities of jumping. We do not rely on prior transition probabilities $P_{A,S}$, but rather optimize the action taking using the observation likelihoods $p\left(d_{j}\left|{\hat{k}}_{i}\right.\right)$ through our track confidence score χ. The motivation for modeling the prior jump probabilities with a uniform distribution is because we suspect that all state jumps are governed by the target-data association quality. Therefore, any prior belief in taking a specific action, as modeled in [47], has the potential disadvantage to discriminate and impose bias in the decision process. In other words, current dataset sizes do not provide enough evidence for accurate modeling of prior jump probabilities, so any learned prior will inherently over-fit to the specific scenarios in the dataset.

By enforcing the Markovian principle, we designed a set of actions A and rewards *R* inspired by the MDP jump states in [47] and the Reversible Jump moves in [33]. These actions bring the target status in a dynamic equilibrium using the confidence hysteresis property of χ:

**Update: $A_{update}$** defines the action of incorporating new observation data $d_{j}$ to tracked target $k_{i}$ after a successful association, $\left(i,j\right)\in L$. Formally, the transition: (47)$$T:\left(S_{k_{}}^{\left\{\mathrm{active},\mathrm{confident}\right\}},d_{j}\right)\times A_{update}\mapsto S_{k_{}}^{\left\{\mathrm{active},\mathrm{confident}\right\}}$$ moves our target from the active to the confident state, or vice versa based on the newly computed confidence score $\chi^{t}$. The reward function $R_{update}()$ is simply the newly computed $\chi_{t}$ value:(48)$$R_{update}\left(\left(S_{k_{}}^{\left\{\mathrm{active},\mathrm{confident}\right\}},d_{j}\right)\times A_{update}\right)=\chi_{i}^{t}.$$

This action allows for the weight update of the PF to be performed and thus the following re-sampling and MAP computation. Contrary to [33] where the update action is integrated as a proposal function in the particle filter itself, here we formulate updating as a high-level requirement for updating the PF. Namely, for each association tuple $a_{i,j}$ satisfying Equation (Equation 39) ($\left(i,j\right)\mathrm{in}L$) we assign observation $d_{j}$ to the PF of target $k_{i},$ and compute the reward function Equation (Equation 48). This reward function defines the importance of assigning data to a target at time *t* and can thus increase our confidence in the target. The result of the update action, as seen in Equation (Equation 47), can also keep a target in the active state Figure 7. This can happen whenever $\chi_{t}<thr_{conf}$.

**Predict: $A_{predict}$** defines the default action taken for all tracked targets before observing and associating with new detections. The predict action is formulated as the empty update using the state transition prior: (49)$$T:\left(S_{k_{}^{t-1}}^{\left\{\mathrm{active},\mathrm{confident}\right\}},\emptyset \right)\times A_{predict}\mapsto S_{k_{}^{t}}^{\left\{\mathrm{active},\mathrm{confident}\right\}},$$ where we decrease the predicted target confidence value as in Equation (Equation 45). The reward function for the predict returns the newly computed χ and governs the jump into an active, if $S_{k}^{active}:\chi_{t}<thr_{conf}$, or confident state otherwise:(50)$$R_{predict}\left(\left(S_{k_{}}^{\left\{\mathrm{active},\mathrm{confident}\right\}},\emptyset \right)\times A_{predict}\right)=\chi_{t}.$$

A “non-assigned” confident track can stay up to several time steps in the confident state; this happens for targets that quickly disappear from view of the detector. Such targets can be reliably tracked based on the state transition prior, $p\left(k_{t}\left|k_{t-1}\right.\right),$ and the learned PF distribution, $p\left(k_{t-1}\left|d_{t-1}\right.\right),$ alone. Once the confidence value drops below the confident threshold, a counter measures the time the track has been taking only the predict action. After $N_{predict}$ time steps, active targets can trigger the remove action.

**Remove:**$A_{remove}$ is taken by active tracks that are not associated with new observations over prolonged periods of time. Such tracks are no longer actively tracked and our confidence in the state variables is close to zero, χ≈0. Formally, a track is removed, i.e., it becomes lost, when it can not be consecutively associated with a new observations, $\forall \left(i,j\right)\notin L$, after a number of time steps $N_{predict}$:(51)$$T:S_{k_{}}^{\mathrm{active}}\times A_{remove}\mapsto S_{k_{}}^{\mathrm{lost}}.$$

The reward function for removing a target $R_{remove}$ is defined as:(52)$$R_{remove}\left(S_{k_{}}^{\mathrm{active}}\times A_{remove}\right)=\left\{\begin{array}{cc} 1 & \mathrm{if}\chi_{t}\approx 0,\\ 0 & \mathrm{otherwise}. \end{array}\right.$$

This part of our track management is completely deterministic and cleans up tracks with low confidence scores after a certain time period of non-assignment. After removal, the track confidence score of a lost target is χ=0. A removed/lost target will not update itself with the state transition prior and can only be re-identified if a suitable detection becomes available.

**Add: $A_{add}$** defines the opposite action of $A_{remove}$ when a new observation $d_{j}$ can not be associated with the current tracks and carries strong enough observational evidence that it can start its own track, Figure 6 green observation: (53)$$T:\left(S_{k_{}}^{\mathrm{inactive}},d_{j}\right)\times A_{add}\mapsto S_{k_{m+1}^{}}^{\left\{\mathrm{active},\mathrm{inactive}\right\}}._{}$$

This action creates a new track **$k_{m+1}$** from an unmatched observation **$d_{j}$** with a high detection score $s_{j}$, i.e., all associations $\left(i,j\right)$ that are not part of the optimal solution L in Equation (Equation 39). Formally, the add action is governed by the reward function $R_{add}()$:(54)$$R_{add}\left(\left(S_{k_{}}^{\mathrm{inactive}},d_{j}\right)\times A_{add}\right)=\left\{\begin{array}{cc} s_{j} & \mathrm{if}s_{j}>0.5\wedge \left(\forall \left(i,j\right)\notin L\right),\\ 0 & \mathrm{otherwise}. \end{array}\right.$$

This means that unassociated observations with high detection scores are more likely to spawn an active track, and, otherwise, these observations are discarded and no new track is created. The track confidence score for a new target is initialized as $\chi_{m+1}^{0}=1$.

**Merge: $A_{merge}$** merges confident targets that become too similar after an update: (55)$$T:\left(S_{k_{i}}^{\mathrm{confident}},S_{k_{j}}^{\mathrm{confident}}\right)\times A_{merge}\mapsto \left\{\begin{array}{cc} S_{k_{i}}^{\mathrm{confident}}, & \mathrm{if}\chi_{i}\geq \chi_{j},\\ \;S_{k_{{}^{j}}}^{\mathrm{confident}}, & \mathrm{otherwise}, \end{array}\right.$$ where, based on their confidence scores $\chi_{i},\chi_{j},$ the one with the higher confidence score is more likely to survive while the other one is more likely to becomes inactive. This is an important track to track interaction behavior which controls the formation of false positive trajectories around a single pedestrian. Since we only merge confident tracks, it is possible that an active track occupies the same physical area with another active track. The mechanism, however, is in line with the multiple hypothesis paradigm and helps the tracker in ambiguous situations where we want to keep track of multiple possible trajectories with lower confidences. We do the merging based on a similarity metric of two targets in image and ground plane distances. Hence, the reward function for merging two targets is defined as:(56)$$R_{merge}\left(\left(S_{k_{i}}^{\mathrm{confident}},S_{k_{j}}^{\mathrm{confident}}\right)\times A_{merge}\right)=\left\{\begin{array}{cc} f_{merge}\left(k_{i},k_{j}\right) & \mathrm{if}\chi_{i},\chi_{j}\geq thr_{conf},\\ 0 & \mathrm{otherwise}, \end{array}\right.$$
(57)$$f_{merge}\left(k_{i},k_{j}\right)=\mathrm{IoU}\left({\hat{k}}_{i},{\hat{k}}_{j}\right)\cdot \exp\left(-\frac{d_{3D}{\left({\hat{k}}_{i},{\hat{k}}_{j}\right)}^{2}}{\sigma_{merge}^{2}}\right),$$ where IoU computes the Jaccard Index (Intersection over Union) of the two image plane bounding boxes. The reward of merging two targets becomes large when the two are close together in the combined 2D–3D space.

**Re-identify:**$A_{reID}$ is taken whenever an observation $d_{j}$ that can not be associated with any active target is similar in appearance to one of the lost targets:(58)$$T:\left(S_{k_{i}}^{\mathrm{lost}},d_{j}\right)\times A_{reID}\mapsto S_{k_{i}}^{\left\{\mathrm{active},\mathrm{lost}\right\}},$$ where the re-identification reward function $R_{reID}$ is defined as:(59)$$\begin{array}{c} R_{reID}\left(\left(S_{k_{i}}^{\mathrm{lost}},d_{j}\right)\times A_{reID}\right)=\\ \left\{\begin{array}{cc} f_{reID}\left(d_{j},k_{i}^{\mathrm{lost}}\right) & \mathrm{if}s_{j}\leq 0.5\wedge \left(\forall \left(i,j\right)\notin L\right),\\ 0 & \mathrm{otherwise}, \end{array}\right. \end{array}$$ and $f_{reID}$ consists of the observation likelihood, Section 5.3, in 3D and Appearance space:(60)$$f_{reID}\left(d_{j},k_{i}^{\mathrm{lost}}\right)=f_{1}\left({\hat{k}}_{i},d_{j}\right)\cdot f_{4}\left({\hat{k}}_{i},d_{j}\right).$$

Finally, the target confidence score χ of a re-identified target is re-initialized. The proposed track confidence score and management brings the system to a state of dynamic equilibrium between true observations and tracklets which is the fundamental multi-object tracking property that we seek to achieve.

## 7. Evaluation and Results

### 7.1. Evaluation Metrics

We evaluate the performance of our tracker using the MOT16 benchmark method proposed in [48]. This methodology combines both quantitative, CLEAR metric [49], and qualitative, Track Quality Measure [50], tests of tracker performance. Since our system builds upon a standard object detection algorithm we will not benchmark the actual detection performance which is given. We will herein shortly discuss the evaluation metrics, however, the reader is advised to refer to the work in [48] for more comprehensive elaboration and underlying reasoning.

**1. Tracker-to-target assignment:** This metric describes the reliability of the Multi-Object Tracker by measuring the number of False Positive (FP) and False Negative (FN) per and across all frames given by the **F1 score**. Tracker robustness is also tested by measuring the number of fragmentations (i.e., losing the track and starting a new one) for the same tracked object. This is indicated by the number of tracker ID switch (IDS) across all frames. An ideal tracker would improve upon the FP and FN rates of the object detector while at the same time have a fragmentation or ID switch score of zero.

**2. Multiple Object Tracking Accuracy** (MOTA): combines FP, FN and IDS to indicate the overall performance of the tracker. Formally, **MOTA** is the ratio:(61)$$MOTA=1-\frac{\sum_{t}\left(FN_{t}+FP_{t}+IDSW_{t}\right)}{\sum_{t}GT_{t}},$$ where *t* is the time step (frame index) and GT is the number of Ground Truth objects. The value of MOTA can also be negative if the number of errors exceeds the number of actual objects. Most trackers in the literature are compared primarily using this metric since it represents a good balance between tracking precision, recall, and temporal stability.

**3. Multiple Object Tracking Precision** (MOTP): is the total position error for matched object-hypothesis pairs over all frames, averaged by the total number of matches made. It shows the ability of the tracker to estimate precise object positions, independent from its skill at recognizing object configurations, keeping consistent trajectories, etc. Essentially, it indicates the local accuracy of the tracker. **MOTP** is computed as:(62)$$MOTP=\frac{\sum_{t,i}d_{t,i}}{\sum_{t}c_{t}},$$ where $c_{t}$ denotes the amount of tracker-target matches in frame *t* and $d_{t,i}$ is the bounding box overlap between tracked target *i* with the GT. The overlap is measured by means of intersection over union (IOU).

**4. Track quality measure:** This measure classifies the tracking output based on how many trajectories of the GT it covers. Mostly Tracked (**MT**) corresponds to at least 80% coverage, and Mostly Lost (**ML**) means that the track is only covered for only less than 20%. Another indicator is Fragmentation Number (**FRAG**), which is the number of a track interruption before it resumes the previously lost trajectory.

### 7.2. Dataset and Implementation Details

To this date, the KITTI tracking dataset [1] provides the most realistic benchmark for 2D–3D object tracking in an automotive setting. This public dataset provides raw RGB camera frames recorded at regular time intervals of 100 ms. Intrinsic camera calibration parameters are available together with distortion coefficients for correcting the lens degradation. Additionally, the dataset also contains raw point cloud data recorded with a 64 beam 3D LiDAR which is calibrated and time synchronized with the camera. The recordings are made in a diverse urban driving scenario recorded on the city of Karlsruhe, Germany. The recording also captures real-world traffic situation and range from highways over rural areas to inner-city scenes. Ground truth is provided by means of high-quality hand-labelled annotations of objects in the camera image containing information such as object class, position, size, orientation and identity. The LiDAR sensor is a Velodyne HDL-64E unit (San Jose, CA, USA) while the cameras are Pointgrey Flea (Wilsonville, OR, USA).

The KITTI dataset also provides a tracking benchmarking platform which shares its methodologies with MOT16 where performance is evaluated primarily for tracking using camera data, with LIDAR data synchronized to the camera image. Therefore, a LIDAR object is only annotated if and only if the object is within camera frame. There are naturally numerous frames when the object is visible on camera but completely missed by LIDAR sensor (i.e., out of range). The opposite is also true: the LIDAR is 360 degrees surround sensor while the camera is a single view frontal view sensor; therefore, objects that are located on the sides and back of the ego vehicle will not have corresponding LIDAR ground truth. Moreover, the Velodyne range data is not perfect representation of the environment correlating with object reflectivity, with a quoted range of 50 m (10% reflectivity) and 120 m (80% reflectivity). In practice, we found that the effective range of the sensor is limited to 40 m. Beyond this range, the number of points reflected is very low and the object shape is not easily recognized even to a trained human eye. The tracker might still be capable of tracking the position and predict the trajectory based on image data, but no useful ground plane information can be derived until such object comes into the sensor effective range.

The dataset is split into two disjoint sets of video sequences, 29 of which are used for independent evaluation at the server side and 21 sequences have ground truth available for training. A total of 172 unique pedestrians are labeled with 11,470 instances across the 8008 training frames.

As mentioned in Section 5.1, we base our particle filter motion model on the behavior of annotated pedestrians from training data. For this, we exploit the 21 KITTI training sequences where consecutive pedestrian motion trajectories are given in the image domain. Using a matching technique for 2D to 3D matching, which is outside of the scope of this paper, we extract the respective 3D trajectories for each annotated pedestrian. From there, by applying a short term temporal analysis of the dynamics of these trajectories, we estimate the short-term behavioral qualities such as probability of walking at a certain pace, probability of acceleration and probability to change the direction of motion. During testing, we apply the SubCNN [40] pedestrian detector to generate observations, i.e., the candidate detections. For transparency, we use the same detections provided by a competing method [47]. Association cost variances in (Equation 30) are empirically chosen such that they equalize the range of association similarity in the different modalities and are $\sigma_{1:5}=\left\{\sqrt{20},\sqrt{5},\sqrt{2},\sqrt{20},\sqrt{2}\right\}$, respectively. This way, our cost function is balanced across each term and no particular constituent metric is given preference.

Tracker logic is based on our MDP state machine with hysteresis temporal confidence scores χ. The optimal action taken for each state is found by a greedy algorithm which selects the action with the highest reward value given the target-data association. Each action leads to exactly one state. When updating χ, we use the null-hypothesis probability C=0.05 and $P_{D}=0.52$ which were emprically chosen as reasonable approximations for the employed object detector performance. We output targets with confidence value scores $\chi >thr_{conf}$ where the confident threshold is adaptive with respect to the ego vehicle velocity (expressed in km/h):(63)$$thr_{conf}={\left[1-\exp\left(\frac{-{\left(v+2\right)}^{2}}{\sigma^{2}}\right)\right]}^{0.1},$$ where the threshold values are lower if the vehicle is static and close to 1 if the vehicle is moving fast. This adaptive threshold is based on the belief that the overall performance of the tracker decreases with the increase of ego velocity, thus the need for higher probability threshold. A target is considered lost if it doesn’t get any observation associated for 50 consecutive frames (5 s). Re-identification is performed for every fresh tracklet that stays unassociated with the currently active tracklets. We do re-identification based solely on the product of ground plane and appearance metrics $f_{1}\cdot f_{4}$. Each tracked pedestrian is represented by the state estimate (mode) of the dual particle filter for which we use 8×1000 particles. Our system takes around 26 ms for the entire prediction, similarity computation, data association and update chain. The system runs on a Linux PC equipped with a six-core CPU i7-4930K, 64 GB main memory and a Geforce TitanX (Pascal) graphics card (Santa Clara, CA, USA). Without taking into account external data sources such as pedestrian detection, optical flow and odometry, our tracker runs online and in real time. A working proof of concept tracking code is written in the Quasar programming language [51].

### 7.3. Performance Evaluation

To evaluate the quantitative and qualitative performance of our tracker, we submitted our outputs to the KITTI tracking evaluation server. This methodology guarantees fair evaluation and comparison to relevant methods where the test data is not known. The evaluation server measures several metrics such as MOTA, MOTP, MT, MT, ML, IDS, FRAG, and several object detection scores such as precision, recall, F1, TP, FP, and FN. Authors also self report their average run-time. Authors are also given limited number of trials before submitting the final results in order to avoid parameter tweaking and learning the test data. Since our method is stochastic in nature, we are not able to submit multiple runs of the tracker to get an average and thus we only submitted the first output of the finalized tracker. These results were already visually reasonable as will be discussed in the next chapter by providing some examples.

In Table 1, we provide a snapshot of the official results table for the KITTI tracking benchmark at the time of submission (19.09.2018) where we compare our results to 20+ trackers. Our tracker outperforms all published trackers with a MOTA score of 0.504, Figure 8 top left. In the MOTP metric, which measures the positional accuracy of tracked pedestrians, we again achieve state-of-the-art performance among all published trackers with a score of 0.728, Figure 8 top right. In total, we report 14,059 true positives with only 2043 false positives and 9207 false negatives. This high score indicates that the proposed method is able to reliably track most of the pedestrians without creating too many false positives. The temporal quality score MT ranks our method as #2 with MT=25.43 only behind [12] which is a “near online” method. We also report the recall rate of our method at 0.604 with a precision of 0.873 which gives an F1 score of 0.714. The number of identity switches for the KITTI test benchmark is 235 with a total of 967 fragmentations.

We suspect that the MOTA score relies heavily on the quality of the input object detections. Thus, using a pedestrian detector with high recall and precision rates can potentially yield a tracker with higher MOTA. Our proposed tracker uses the same bounding boxes as [47] and, by exploiting the depth information from LiDAR point clouds in a 2D–3D particle filtering tracker, we are able to significantly improve upon the state-of-the art. Our method does the learning of motion behavior parameters offline and has a much faster execution time. However, as seen on the results page, there exist trackers with an unknown source that produce even higher MOTA scores. This can be due to many factors that are currently unknown and can not be objectively compared.

One last remark is our high MOTP score which is irrelevant of precision/recall curves of object detections and only measures the positional accuracy of the pedestrians that are tracked. This finding best reinforces our choice to use particle filters that have the freedom of representing object positions as a nonparametric distributions. By using our behavioral priors in the particle motion updates, we can better predict future positions and better associate new detections with existing tracklets.

### 7.4. Qualitative Analysis and Discussion

We conclude our analysis with several qualitative tests that represent the border cases of our tracker, i.e., we show several difficult scenarios where tracking is performed with high accuracy and several cases where it fails. In a classical multi-object tracking setup, most of the errors come when objects interact with the background or with each other. When a person walks past another person or an occluding object, parts of him/her become non-visible. Candidate object detections in such cases might fail completely or become less confident. This, in turn, causes problems in the data association where cost functions become ambiguous and matching becomes false. Our tracker tackles these problems using depth and azimuth information. In our experiments, border cases with difficult occlusions and sudden appearance changes become trivial. In Figure 9, we present two cases of difficult occlusion where our method is able to continue tracking once the respective person re-appears without any loss of precision in the meantime.

In Figure 9A, the EGO vehicle is standing still on a pedestrian crossing and groups of people are walking in opposite directions. This sequence is particularly demanding since several occlusions happen in the middle portion that is covered by a shadow. Here, appearance models change quickly and can not be updated due to occlusion. We observe that our tracker is able to keep track of every person regardless of occlusion which we suspect is possible only because we use additional depth and azimuth information in our particle filter. Once the tracked person becomes occluded, the particles continue to be updated based on the prior behavioral motion. When the person re-appears again, his/hers expected 3D position can be accurately matched to new measurements.

The scenario in Figure 9B presents a situation where the EGO vehicle is moving and detected pedestrians to the left are occluding more distant pedestrians in the background. Due to the relatively low frame rate, these occlusions happen in quick succession and the background person is only detected every other frame. However, from the images shown on the figure, it is visible that our tracker can estimate the position of the occluded person and easily re-engage tracking when fresh detections appear. In this scenario, much of the tracking information comes from the optical flow and odometry estimation. By knowing the distance of the person prior to his/hers occlusion, the tracker can estimate the optical flow field at that distance and continue to adjust the particle positions accordingly. This way, when the person re-appears, the expected location closely matches new detections.

Lastly, the three frames in Figure 9C visualize some of the fail cases of our method. Here, highly confident tracklets appear in positions where there is a random object with a pedestrian like appearance. Since our tracker is based on detection and runs online, it makes the most informed decision in the present. Such tracking is not optimal since tracked objects can be far away and detection confidence might, at the present time, be low. Once the EGO vehicle drives close to these objects, their appearance improves and the detector disregards them as background objects. In the literature, such ambiguities are easily solved offline by forward-backward error validation; however, the problem still remains one of the fundamental issues in detection based online systems.

## 8. Conclusions

There exist various complex offline methods which operate at high accuracies, but accurate online tracking still remains an open research topic. Motivated by handling the difficult cases of tracking under bad lighting and occlusion, in this paper, we propose a system for online tracking of multiple pedestrians from a moving vehicle. Observational data is provided in the visible light and 3D domain by a weakly calibrated camera-LiDAR pair, where classical computer vision techniques are run in order to extract potential objects of interest for tracking. Based on the principles of tracking by detection and prediction-update cycles, our system fits in a standard perception pipeline where other modular blocks such as object detectors, optical flow, odometry, etc. can be added or interchanged in the future. The tracker applies non-parametric distributions for state estimation and exploits offline knowledge of pedestrian behavior to model unpredictable motion. We showed that, by running multiple Monte Carlo simulations, the tracker is able to accurately track pedestrians in spite of heavy occlusion and bad lighting conditions. This is partly a result of the incorporation of 3D LiDAR measurements into the data association function, resulting in a notable increase in robustness to person-to-person and person-to-background occlusions. To this date, we can report that we achieve state-of-the-art tracking performance among published trackers on the KITTI benchmark using a publicly available object detector as input.

Still, some of the results remain open to interpretation and, as such, are the topic of our ongoing research. First, quantitative evaluations show that our tracklets have many identity switches and fragmentations. This is in part due to the online nature of operation and the simplistic appearance model used for re-identification. More research is needed, especially in proposing a temporarily stable motion model that explains long-term pedestrian behavior. In our future work, we intend to examine the effects of person to person interactions and group forming behavior. Another interesting behavioral analysis can be performed if information on the traffic infrastructure is also available to the tracker. Long-term intentions of pedestrians, such as: crossing on a green light or giving the right of way to other users can be better incorporated into the positional updates. Second, our quantitative analysis mainly focuses on labeled image plane data. This is in turn tied to being able to objectively compare to competing methods and the availability of evaluation datasets. As can be seen on our online project page, our tracker also outputs high-quality ground plane trajectories. The accuracy of these trajectories remains unmeasured since no such benchmark currently exists in the literature. Lastly, there is a need for analysis of how tracking performance scales with different combinations of image and depth sensors. We are particularly interested in deploying the tracker in a realistic low-cost hardware setup where depth sensing will be performed by a low-cost LiDAR or 2D RADAR. Our ongoing research includes the creation of such platform and building a realistic dataset for evaluation.

## Figures and Tables

**Figure 1 sensors-19-00391-f001:**
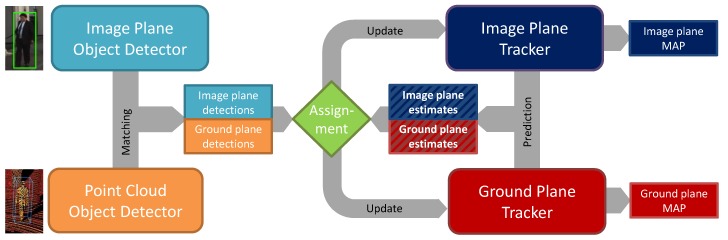
A simplified schematic depicting the various paths of information flow and feedback between sub-components in the proposed system. **Left**: CNN detections, in blue, are matched to 3D segments originating from the LiDAR, in orange. **Right**: image plane tracker, shown in blue, and ground plane tracker, in red, form predictions and updates. **Center**: matching of observations and track estimates are done by optimizing the joint observation log-likelihood, in green.

**Figure 2 sensors-19-00391-f002:**
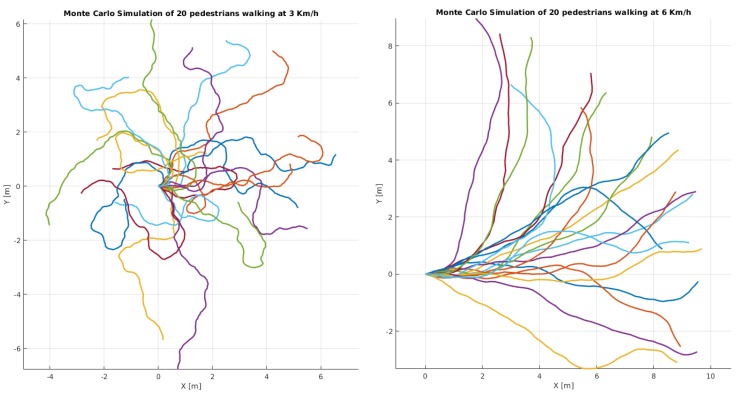
Simulated pedestrian motion on the ground plane using our behavior motion priors learned from the KITTI tracking dataset. **Left**: 20 particles starting from 3 km/h; **Right**: 20 particles starting from 6 km/h.

**Figure 3 sensors-19-00391-f003:**
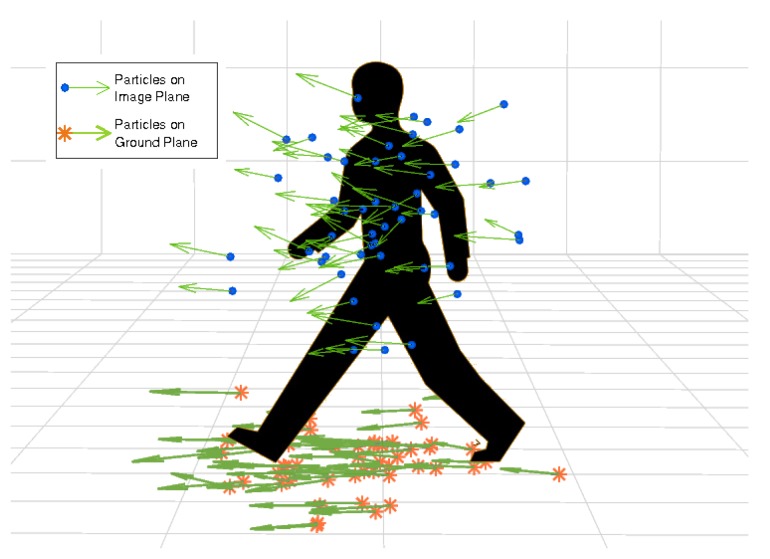
Graphical illustration of the layout of our 2D–3D particle filter tracker. State variables on the image plane are modeled by particles (blue) which are independent from the particles that model the ground plane state variables (orange).

**Figure 4 sensors-19-00391-f004:**
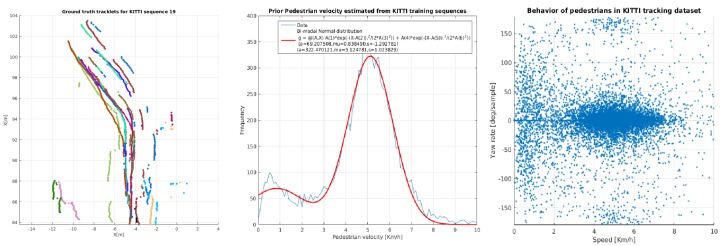
Example of the ground truth data used to model our behavioral motion priors. **Left**: ground truth tracklets from the 0019 tracking sequence in the KITTI tracking dataset. **Middle**: Estimated prior probability distribution of walking pace. **Right**: Measurements of yaw rate relative to the walking pace.

**Figure 5 sensors-19-00391-f005:**
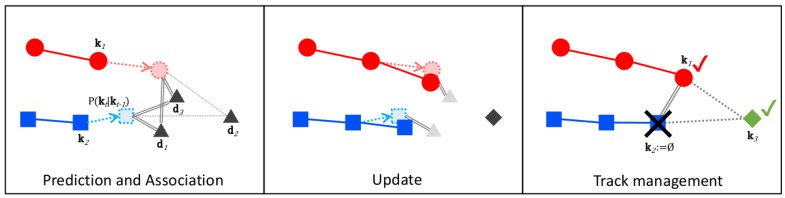
Illustration of the employed divide and conquer strategy. **1st step:** predictions based on motion priors (dashed lines) are associated with observations (triangles). **2nd step:** feasible tuples (double lines) are updated using the track management policy and, at the same time, other proposal moves such as adding or merging are also considered. e.g., on the right $T_{1}:\left\{S_{blue},S_{red}\right\}\times A_{merge}\mapsto S_{red}$, $T_{2}:A_{add}\left(d_{j}\right)\mapsto S_{green}.$

**Figure 6 sensors-19-00391-f006:**
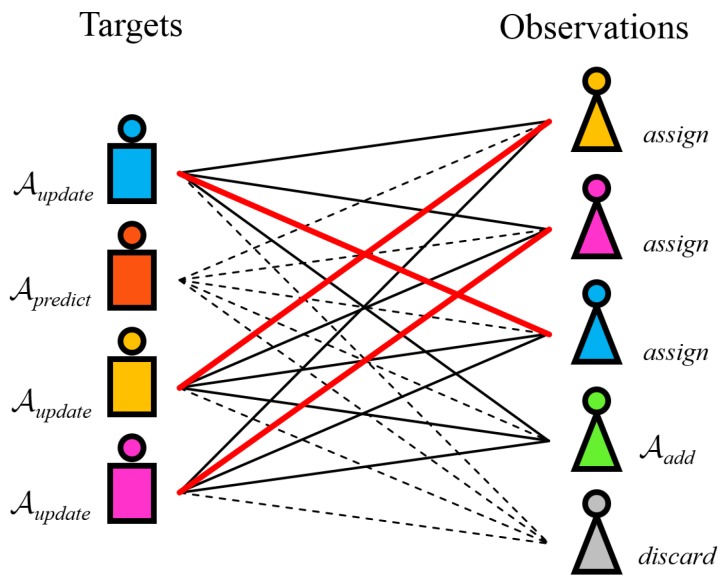
An example association graph between four targets and five detections. Optimal associations are shown as red edges, while the infeasible ones are dashed. One target can not be matched to any detection, and two detections can not be matched to any target. The respective actions taken by the tracker are presented next to each vertex.

**Figure 7 sensors-19-00391-f007:**
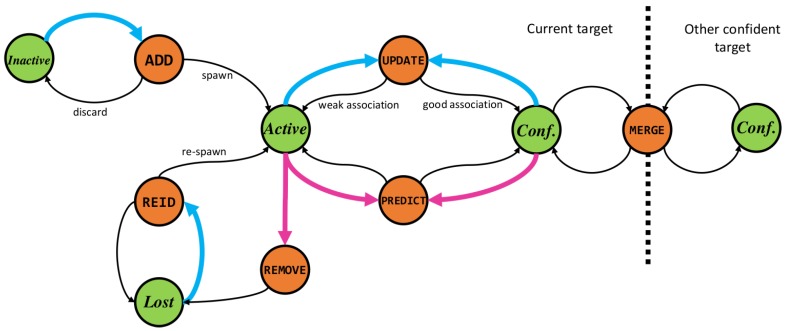
State diagram of a track life cycle. A track can change its state (green) by initiating different Actions (orange) based on the observation to target association solution L. Blue arrows indicate actions where observation can be associated, while magenta arrows indicate action taking with no available observation.

**Figure 8 sensors-19-00391-f008:**
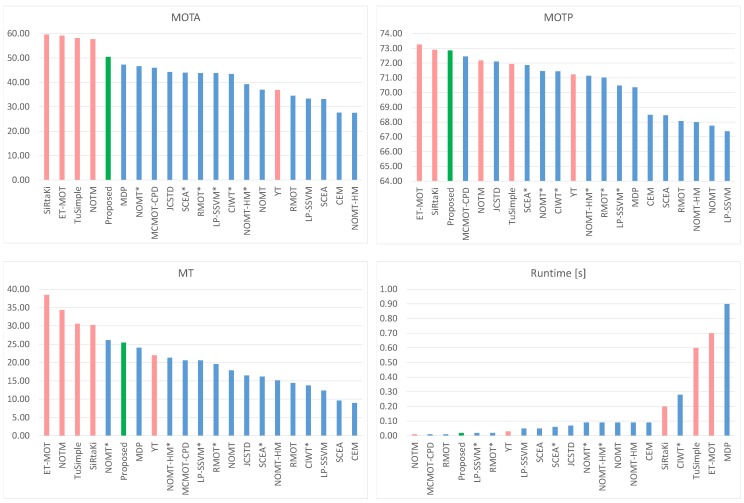
Comparison of the proposed method with various evaluated Pedestrian trackers on the KITTI tracking test set. Red indicates trackers with unknown source, blue bars are trackers published in the literature and green bars are results from the proposed tracker. Higher MOTA, MOTP and MT scores are better.

**Figure 9 sensors-19-00391-f009:**
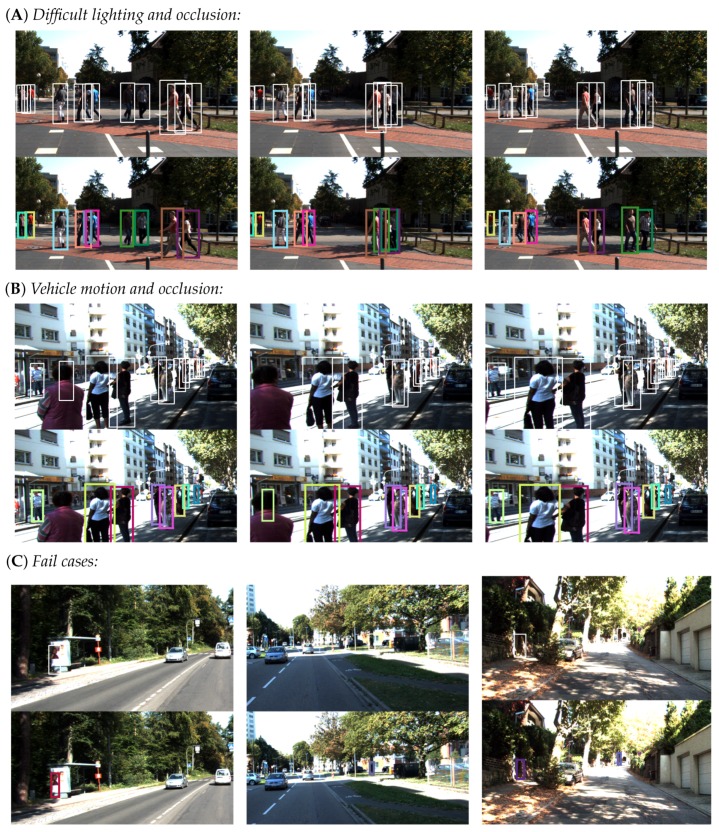
Examples of our tracking output in typical situations that are considered difficult in a classical tracking setup. Frames with raw pedestrian detections are shown on top of tracked pedestrians for the respective sequence. In (**A**), the focus is on the pair of people (green tracklets) passing with the other pair of people and coming out from a tree shadow. While their appearance changes and there is a complete occlusion, they are effectively tracked. In (**B**), the focus is on the person across the street who gets occluded due to the motion. Our method is able to compensate for this motion and continue tracking when he becomes visible again. In (**C**), we show three situations of persistent clutter where our method is tracking random objects that resemble pedestrians.

**Table 1 sensors-19-00391-t001:** Results on KITTI pedestrian tracking dataset sorted by MOTA score. Setting on = Online, la = Laser Data, st = Stereo Data. Red rows belong to submissions from unknown sources.

Method	Setting	MOTA	MOTP	MT	ML	IDS	FRAG	Runtime	Source
SiRtaKi	on	59.61%	72.89%	30.24%	15.81%	136	1164	0.2 s/GPU	Unknown
ET-MOT	on	59.10%	73.26%	38.49%	10.31%	316	1362	0.7 s/GPU	Unknown
TuSimple	on	58.15%	71.93%	30.58%	24.05%	138	818	0.6 s/1 core	Unknown
NOTM		57.67%	72.17%	34.36%	19.24%	108	799	0.01 s/1 core	Unknown
Be-Track	la on	50.39%	72.85%	25.43%	27.84%	235	967	0.02 s/GPU	PROPOSED
MDP	on	47.22%	70.36%	24.05%	27.84%	87	825	0.9 s/8 cores	[47]
NOMT *		46.62%	71.45%	26.12%	34.02%	63	666	0.09 s/16 cores	[12]
MCMOT-CPD		45.94%	72.44%	20.62%	34.36%	143	764	0.01 s/1 core	[52]
JCSTD	on	44.20%	72.09%	16.49%	33.68%	53	917	0.07 s/1 core	[53]
SCEA *	on	43.91%	71.86%	16.15%	43.30%	56	641	0.06 s/1 core	[54]
RMOT *	on	43.77%	71.02%	19.59%	41.24%	153	748	0.02 s/1 core	[55]
LP-SSVM *		43.76%	70.48%	20.62%	34.36%	73	809	0.02 s/1 core	[56]
CIWT *	st on	43.37%	71.44%	13.75%	34.71%	112	901	0.28 s/1 core	[13]
NOMT-HM *	on	39.26%	71.14%	21.31%	41.92%	184	863	0.09 s/8 cores	[12]
NOMT		36.93%	67.75%	17.87%	42.61%	34	789	0.09 s/16 core	[12]
YT	on	36.90%	71.22%	21.99%	25.43%	267	995	0.03 s/4 cores	Unknown
RMOT	on	34.54%	68.06%	14.43%	47.42%	81	685	0.01 s/1 core	[55]
LP-SSVM		33.33%	67.38%	12.37%	45.02%	72	818	0.05 s/1 core	[56]
SCEA	on	33.13%	68.45%	9.62%	46.74%	**16**	717	0.05 s/1 core	[54]
CEM		27.54%	68.48%	8.93%	51.89%	96	608	0.09 s/1 core	[7]
NOMT-HM	on	27.49%	67.99%	15.12%	50.52%	73	732	0.09 s/8 cores	[12]

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
