# Peer review of "Behavioral Pedestrian Tracking Using a Camera and LiDAR Sensors on a Moving Vehicle"

_sensors, 2019, doi:10.3390/s19020391_

Round 1
Reviewer 1 Report
This paper investigates the pedestrian tracking problem using particle filter. In this paper, the target motion model is divided into two parts: the ground plane motion model and image plane model, according to the motion model, the observations which is the union of the ground and image observations are used to update the predicted particles. In the reviewer’s view, the main contributions of this paper are the division of the target behavioral motion model and the data association process.
In the reviewer’s view, this paper is addressing a complex practical problem in target tracking. The results of this paper are also of practical importance. However there are still several important issues needed to be discussed, especially the statement of the proposed 2D-3D MOT particle filter and the observation model.
1) In Section 4, the authors propose the 2D-3D MOT Bayesian tracking and 2D-3D MOT particle filter, However, the proposed Bayesian tracking framework and particle filter confuse the reviewer, in the reviewer’s view, the proposed Bayesian tracking framework is very similar with the standard Bayesian tracking framework and the Bayesian tracking equations derived in this section is the standard Bayesian tracking equations. The proposed 2D-3D MOT particle filter is exactly the bootstrap filter with the proposed target motion models. It will be more interesting if the authors point out the difference between the standard particle filter and the proposed particle filter.
2) In Section 5, the system model and the observation model are introduced, the likelihood function is obtained in equation (31), while the reviewer wonders what’s the meaning of the notion “”. Assume that the notion “” means the matrix norm and the observations are modeled as Gaussian distribution in this literature, may be the projection function is more suitable than the state vector in (31) and the following.
3) In this paper, the ground observations and the image observations are both obtained to update the target states. In Section 3, the authors mention that the observation vector consists of the image data obtained by camera and ground plane location data obtained by LiDAR sensor respectively, the image data and the ground data have been matched before they can be used in tracking process. The reviewer notices that few words are used to describe the data matching process and wonders how the authors matching these two kinds of data.
4) Usually, there is no data association process in standard particle filter, the performance of multi-target particle filter suffers from the interaction of targets in proximity. In the reviewer’s opinion, the authors solve this problem by using the CNN detection and data association process, thus the data association is necessary in the particle filter used in this paper. The authors should elaborate more clearly the importance and the benefits of the data association, clearly showing the importance of data association can also better motivate the proposed method in this paper.
Finally, the English writing of this paper haven’t reach the standard of publication. Careful proofreading and revisions are required. There exist many errors and typos in the paper. Some instances are given as follow.
-Page 10, equation (4),a notion is missed in this equation.
-Page 10, Line 359, “ad” should be “and”.
-Page 10, Line 351 and Line 362, there should be no indentation before “where” and “or”, similar mistakes should be corrected.
-Page 27, Line 814, “3D matching described in section ?? we extract the respective 3D trajectories”, the number of section is missed.

Author Response
1) Added a small paragraph pointing out the novelties at the beginning of the section. (Usage of standard bootstrap PF equations with novel behavioral motion model.)
2) In section 5, the likelihood function compares observations to the state estimate projection onto the observation space. E.g. detected bounding boxes on the image plane by the CNN are compared to projected bounding boxes of the tracked person in the image plane, eq (31,32). For the sake of simplicity the projection operator (function f_k()) is omitted in equations 31-37.
3) The matching of 2D image objects to 3D lidar objects is indeed not described in detail as it is not a critical component of the system and no novelties are introduced in this step. In reply to the reviewer: 2D-3D matching is done by directly projecting the point-cloud onto the image plane (a depth image) and then searching for prominent objects in the depth image that match to 2D bounding boxes.
(objects that are distinct from the background are clearly visible and easily segmentable in the depth image)
4) Introductory paragraph of section 6 has been re-worked to put emphasis on why the proposed divide and conquer strategy is effective for real-time multi-object tracking.
-Thorough proof reading was conducted fixing miss-spellings and indentation errors.
-Fixed missing section number. This section was previously a part of the paper, but was later removed for brevity.
Reviewer 2 Report
The issue in the paper are one of the key challenges in AGV and the experimental results obtained have some potential in applications. The comments are as the following:
1.The pedestrian recognition is mainly rely on image processing. However, there are many CNN networks to solve the problem, what is your superiority?
2.Does the matching between the CNN detections and 3D segments originating from the LiDAR rely on the external parameter calibration? As the different position have different calibration precision in a common view, how do you make sure the accuracy of the assignment?
3.It seems that all the techniques used to solve the problems have been proposed in literature. What are the difficultis when dealing with these problems?
Author Response
1) The initial pedestrian candidates are indeed detected by a CNN (presented in section 3.2), namely this method:
"Subcategory-Aware Convolutional Neural Networks for Object Proposals and Detection",Xiang, Y.; Choi, W.; Lin, Y.; Savarese, S. 2017 IEEE Winter Conference on Applications of Computer Vision
The superiority of the tracking system lies in the capacity to handle temporarily unstable detections and ambiguities in person identity due to occlusions and person to person interactions.
With respect to end-to-end tracking CNNs, our system is modular and easier to interpret. At the same time, tracking is performed using realistic likelihood and transition models, which although learned from the annotated data, are general and not very specific to the data-set.
2) Matching of CNN detections and 3D segments assumes that initial calibration of the Camera and LiDAR is performed. This includes extrinsic position and orientation calibration matrices as well as intrinsic camera parameters (focal length, pixel resolution and ratios etc.). Within this paper we exploit a, currently unpublished, matching technique which is robust to small calibration errors. In order not to overload the paper text we opted to not include details about the matching, but should be published in a following paper. In short, LiDAR points are projected on the camera image and segmentation is performed based on the depth of each image pixel. Then, the dominant segment within each CNN detection is considered as a match and distance and azimuth are calculated based on re-projecting this segment into the point cloud. The technique is robust enough that it currently is not the bottleneck for performance.
3. Our system builds upon well established methods from the literature. However, there exist no single theoretical framework that can track multiple objects both accurately and in real-time.
In order to track unpredictable pedestrian behavior, we resort to particle filters which explore the state space in a non-parametric fashion.
Firstly, when using particle filters problems arise if there are many observations and many objects to track. The association between the two can become intractable when associating each particle with each observation. We therefore solve the association separately from the tracking.
Secondly, on-line and real-time requirements force us to rely heavily on current observations and estimates and not so much on historical data. This can have potential issues if the model uses simplistic transition functions, e.g. in the case of occlusion a person might not be visible for a longer period and their position unclear, so many possibilities must be explored. Our state representation PDF (particle filter) is multi modal and allows for multiple local peaks of probable pedestrian position to exist at the same time. The behavioral motion models applied in the paper allow each particle to follow a realistic pedestrian trajectory, so that matching with observations becomes less ambiguous.
Lastly, many complex scenes are simply too difficult to be solved using a single camera sensor. This problem is apparent in crowded streets or in situations where the camera sensor SNR is low. The inclusion of a secondary active sensor (LiDAR) through data fusion brings tremendous robustness to the system since it senses in 3D equally accurate in both good and bad lighting environments.
Round 2
Reviewer 1 Report
In the author’s notes, the questions of the reviewer have been well explained, and thorough proofreading was conducted.
Reviewer 2 Report
This paper presents a method to track the pedestrian by the fusion of the camera and LiDAR data via convolutional neural networks. The experiments show the high performance of the proposed method. If the authors expound how to reduce the matching error in this paper, it will be perfect.
Anyway, the paper is suitable for publication.